# Photoswitching mechanism of a fluorescent protein revealed by time-resolved crystallography and transient absorption spectroscopy

Joyce Woodhouse[1], Gabriela Nass Kovacs[2,12], Nicolas Coquelle [1,3,12], Lucas M. Uriarte [4,12], Virgile Adam [1,12], Thomas R.M. Barends [2], Martin Byrdin [1], Eugenio de la Mora [1], R. Bruce Doak[2], Mikolaj Feliks[5], Martin Field [1,6], Franck Fieschi [1], Virginia Guillon[1], Stefan Jakobs [7], Yasumasa Joti[8], Pauline Macheboeuf [1], Koji Motomura[9], Karol Nass [2], Shigeki Owada[10], Christopher M. Roome[2], Cyril Ruckebusch [4], Giorgio Schirò[1], Robert L. Shoeman [2], Michel Thepaut[1], Tadashi Togashi[8], Kensuke Tono[8], Makina Yabashi [10], Marco Cammarata [11], Lutz Foucar[2], Dominique Bourgeois [1], Michel Sliwa [4*], Jacques-Philippe Colletier[1], Ilme Schlichting [2*] & Martin Weik [1*]

Reversibly switchable fluorescent proteins (RSFPs) serve as markers in advanced fluorescence imaging. Photoswitching from a non-fluorescent off-state to a fluorescent on-state involves *trans*-to-*cis* chromophore isomerization and proton transfer. Whereas excited-state events on the ps timescale have been structurally characterized, conformational changes on slower timescales remain elusive. Here we describe the off-to-on photoswitching mechanism in the RSFP rsEGFP2 by using a combination of time-resolved serial crystallography at an X-ray free-electron laser and ns-resolved pump–probe UV-visible spectroscopy. Ten ns after photoexcitation, the crystal structure features a chromophore that isomerized from *trans* to *cis* but the surrounding pocket features conformational differences compared to the final on-state. Spectroscopy identifies the chromophore in this ground-state photo-intermediate as being protonated. Deprotonation then occurs on the µs timescale and correlates with a conformational change of the conserved neighbouring histidine. Together with a previous excited-state study, our data allow establishing a detailed mechanism of off-to-on photoswitching in rsEGFP2.

[1] Univ. Grenoble Alpes, CEA, CNRS, Institut de Biologie Structurale, F-38000 Grenoble, France. [2] Max-Planck-Institut für medizinische Forschung, Jahnstrasse 29, 69120 Heidelberg, Germany. [3] Large-Scale Structures Group, Institut Laue Langevin, 71, avenue des Martyrs, 38042, Grenoble cedex 9, France. [4] Univ. Lille, CNRS, UMR 8516, LASIR, Laboratoire de Spectrochimie Infrarouge et Raman, F59 000 Lille, France. [5] Department of Chemistry, University of Southern California, Los Angeles, USA. [6] Laboratoire Chimie et Biologie des Métaux, BIG, CEA-Grenoble, Grenoble, France. [7] Department of NanoBiophotonics, Max Planck Institute for Biophysical Chemistry, Göttingen, Germany. [8] Japan Synchrotron Radiation Research Institute, 1-1-1 Kouto, Sayo-cho, Sayo-gun, Hyogo 679-5198, Japan. [9] Institute of Multidisciplinary Research for Advanced Materials, Tohoku University, Sendai 980-8577, Japan. [10] RIKEN SPring-8 Center, Sayo, Japan. [11] Department of Physics, UMR UR1-CNRS 6251, University of Rennes 1, Rennes, France. [12]These authors contributed equally: Gabriela Nass-Kovacs, Nicolas Coquelle, Lucas M. Uriarte, Virgile Adam. *email: michel.sliwa@univ-lille.fr; ilme.schlichting@mpimf-heidelberg.mpg.de; weik@ibs.fr

Phototransformable fluorescent proteins (PTFPs) are invaluable tools for advanced fluorescence microscopy, serving as genetically encoded markers that change emission color or intensity when irradiated with visible light at specific wavelengths[1,2]. A subgroup of PTFPs, the so-called reversibly switchable fluorescent proteins (RSFPs), are used in ensemble-based nanoscopy approaches such as RESOLFT (reversible saturable optical fluorescence transition[3]) or NL-SIM (nonlinear structured illumination microscopy[4]), where they are repeatedly toggled back and forth between a fluorescent on- and a non-fluorescent off-state by irradiation with light at two different wavelengths[5]. Depending on whether irradiation at the excitation peak turns the fluorescent state off or on, RSFP are coined negative or positive switchers, respectively. It is widely accepted that the molecular basis of photoswitching in RSFPs (with the exception of some engineered variants such as Dreiklang[6]) is a combination of isomerization and change in protonation state of the chromophore phenol moiety, as evidenced by X-ray crystallography[7] and absorption and fluorescence spectroscopies[8]. In all negative RSFP, the fluorescent on-state chromophore is an anionic *cis* isomer (*cis*-phenolate) whereas the non-fluorescent off-state is a neutral *trans* isomer (*trans*-phenol). The chronological order of isomerization and protonation-changes is being controversially discussed, as well as corresponding timescales on which they occur.

Over the past decade, several spectroscopic investigations focused on Dronpa, a negative RSFP from Anthozoa (e.g. corals)[9]. Due to the thousand-fold higher switching quantum yield (QY) in the off-to-on (0.37) direction as compared to on-to-off (3.2 × $10^{-4}$)[9], mostly off-to-on photoswitching has been studied experimentally. The first investigation of Dronpa by ultrafast optical spectroscopy suggested that the deprotonation of the *trans*-phenol occurs in the excited state on the ps timescale via an excited-state proton transfer (ESPT) mechanism[10]. Later, however, picosecond time-resolved infrared (TR-IR) spectroscopy on Dronpa[11] and its fast-switching M159T mutant[12] indicated that isomerization occurs during picosecond excited-state decay. This has also been suggested by femtosecond UV–visible transient anisotropy absorption spectroscopy, which attributed the first photoproduct to a *cis*-protonated chromophore[13]. The same study also showed that chromophore deprotonation occurs in the

ground state on the microsecond timescale. However, another TR-IR study on the Dronpa-M159T mutant advocated that both isomerization and deprotonation are ground-state processes, attributing the ps spectroscopic changes in the excited state to protein conformational changes priming the chromophore for switching[14]. This view was corroborated by a follow-up TR-IR study on Dronpa-M159T involving isotope labeling[15]. In that study, evidence has been provided for chromophore distortion and excited-state decay on the picosecond timescale, followed by formation of the *cis* protonated chromophore with a 91 ns time constant before the final deprotonation step. Importantly, the structure of the *cis*-protonated switching intermediate in Dronpa has remained elusive.

RSFPs from Hydrozoa (e.g. jellyfish), such as rsEGFP2[16], are less well studied than those from Anthozoa because fewer examples have been identified or generated so far[17]. Whereas both hydrozoan and anthozoan RSFPs have the same overall protein fold and chromophore (4-(p-hydroxybenzylidene)−5-imidazolinone (p-HBI)), structural features in their chromophore pockets differ. For example, the *p*-hydroxyphenyl ring of the on-state chromophore is stabilized in rsEGFP2 by three hydrogen bonds[18] and in Dronpa by π-stacking with the side chain of a histidine residue and by two hydrogen bonds[7]. Therefore, it is not clear yet if the photoswitching mechanisms in hydrozoan and anthozoan RSFPs differ in details or not. Moreover, structures of ground-state intermediates remain to be determined for both hydrozoan and anthozoan RSFPs. Here, we studied off-to-on photoswitching in rsEGFP2 by a combination of time-resolved crystallography and transient absorption spectroscopy.

rsEGFP2 is a bright, photostable, monomeric and photoswitchable variant of the enhanced green fluorescent protein (EGFP). This fast RSFP is extensively used for tagging proteins in mammalian cells for live-cell RESOLFT nanoscopy[16]. Its p-HBI chromophore, formed autocatalytically from the three residues Ala-Tyr-Gly, is carried by an α-helix entrapped within an 11-stranded β-barrel[18] (Fig. 1a). rsEGFP2 remains photoactive in the crystalline state, so that first insights into the structure of its on- and off-state chromophores could be obtained by conventional crystallography[18]. Optical spectroscopy measurements and crystal structures established that in rsEGFP2, as in other RSFPs such as Dronpa[7], asFP595[19], mTFP0.7[20], or IrisFP[21], the resting on-state

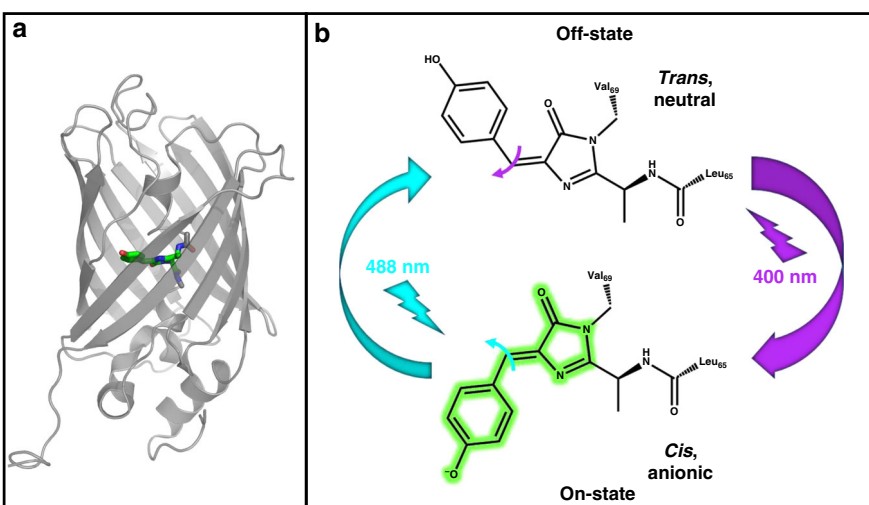

**Fig. 1 Structure and photoswitching scheme of rsEGFP2. a** Three-dimensional structure showing an 11-stranded β-barrel embedding a chromophore held by a central α-helix. **b** The chromophore of rsEGFP2 in its fluorescent on-state (bottom) absorbs at 479 nm (laser 488 nm) and emits at 503 nm. Isomerization competes with fluorescence and leads the chromophore to its non-fluorescent off-state, which absorbs at 403 nm (laser 400 nm) and ultimately isomerizes back to the initial on-state. Flashes and arrows represent the color of each actinic wavelength. The width of arrows is representative of the efficiency of each isomerization.

is characterized by a *cis*-phenolate chromophore, which absorbs at 479 nm and emits at 503 nm (fluorescence QY of 0.35), whereas the off-state features a *trans*-phenol chromophore, which absorbs at 403 nm. Like Dronpa, rsEGFP2 is a negative RSFP, meaning that on-to-off photoswitching competes with fluorescence[18] and is triggered by illumination of the anionic species at 479 nm with a switching $QY^{on-to-off}$ of $1.65 \times 10^{-2}$. Back-switching to the on-state is triggered by illumination at 405 nm, and characterized by a $QY^{off-to-on}$ of 0.33[18] (Fig. 1b).

Chromophore states that accumulate at room temperature and physiological pH can be characterized by conventional X-ray crystallography. In order to characterize the room-temperature structure of photoswitching intermediate states that only exist transiently, another method is needed that allows collecting structural data on a timescale from a few ps to ms. Such a method is time-resolved serial femtosecond (fs) crystallography (TR-SFX) using X-ray free-electron lasers (XFELs)[22,23], which has already provided intermediate-state structures of several photosensitive proteins[24–37]. Recently, we combined ultrafast optical spectroscopy and TR-SFX to study excited-state intermediates of rsEGFP2 on the ps timescale during off-to-on switching[27]. Optical spectroscopy showed that the excited state decays with a time constant of a few ps. TR-SFX revealed that 1 ps after photoexcitation at 400 nm, the chromophore adopts a twisted conformation halfway between the *trans* and *cis* isomers, while at 3 ps, the *cis* chromophore of a presumed ground-state intermediate starts to become populated. The quality of the latter data, however, did not allow building a structural model reliably. Consequently, the structure of the first photoproduct during rsEGFP2 off-to-on photoswitching has remained elusive. Such a structure would provide insight into conformational changes in the chromophore pocket occurring after decay of the excited-state and clarify the order of photoswitching events in RSFPs from Hydrozoa.

Here, we describe TR-SFX experiments that capture an off-to-on switching intermediate, formed 10 ns after photoexcitation of rsEGFP2. The structure of the intermediate reveals that the initially *trans*-protonated chromophore has transitioned to the *cis* isomer and evidences conformational changes in the main and side chains of residues in the chromophore pocket. Structural features revealed by the 10 ns structure are compatible with a *cis* protonated chromophore. Indeed, pump–probe UV–visible spectroscopy indicates that at this time, the chromophore is still protonated, as deprotonation only occurs on the µs to ms timescale in solution. Thus, our data, together with our ps study published earlier[27], establish that off-to-on photoswitching in rsEGFP2 involves excited-state isomerization on the ps timescale followed by µs conformational changes in the ground state that trigger proton transfer on the ms timescale. This allows proposing a detailed switching mechanism.

## Results

**Time-resolved UV–visible transient absorption spectroscopy.** Time-resolved UV–visible transient absorption spectra were recorded from 40-ps to 10-ms after excitation of rsEGFP2 in its off-state in solution (50 mM HEPES pH 8, 50 mM NaCl; Fig. 2, Supplementary Fig. 1). On the examined timescale, the excited state had decayed[27] so that the photodynamics of rsEGFP2 in the ground state was monitored. From 40 ps to 2 ns (Supplementary Fig. 1a), transient spectra show a negative and a positive band at 420 and 460 nm, respectively, evolving with a characteristic time constant of 87 ps ± 8 ps (Supplementary Fig. 1c). A negative band at 500 nm originates from stimulated emission from residual 10% of rsEGFP2 that had remained in the *cis* anionic on-state

(Supplementary Fig. 1b) and vanished after 2 ns (Supplementary Fig. 1d). Between 10 and 100 ns, there is no significant spectral evolution (Supplementary Fig. 2). Figure 2 shows spectral changes from 100 ns to 9 ms. The transient difference absorbance spectrum at 100 ns shows a broad positive band with a maximum at 400 nm (dark blue in Fig. 2a). This band evolves within 4.9 µs to a spectrum (light blue) with two positive maxima at 390 nm and 460 nm. Then within 90.6 µs, the first peak (390 nm) vanishes, while a negative band appears at 410 nm. The second peak (460 nm) increases and shifts to 480 nm (Fig. 2b). Subsequently, and on a timescale from 100 µs to 9 ms, the maximum of the negative band shifts from 420 to 400 nm while the positive band at 480 nm further increases in amplitude. The 480 nm band is characteristic of the *cis* anionic form[16].

Three distinct time windows are observed for the evolution of transient spectra (Fig. 2a–c) with the existence of two isosbestic points (Fig. 2a, c). A model based on the weighted sum of three exponential decays was chosen to fit the kinetic traces for all wavelengths between 100 ns to 10 ms. The estimated time constants are $5.57 \pm 0.02$ µs, $36.1 \pm 0.1$ µs and $824.8 \pm 0.3$ µs, respectively. The residuals show no structure, thus validating the given confidence intervals (Supplementary Fig. 3). However, it is known that standard deviations of the parameter values are optimistic estimates of the confidence on these parameters (since it assumes that the estimated parameters are the true ones)[38]. Therefore, we performed a bootstrap estimate of the standard deviation of the set of parameters. By bootstrapping, different sets of parameters are reported for each replicated fit. Mean value and confidence can then be estimated from their respective distribution. Results obtained for 1000 replicates are reported in Supplementary Fig. 4 (see details in the Methods section) and confirm the validity of the time constants reported above. Similar experiments were carried out in $D_2O$ solution (50 mM HEPES pD 8, 50 mM NaCl; Supplementary Fig. 5), and the same model was applied, yielding time constants of $5.16 \pm 0.02$ µs, $88.4 \pm 0.2$ µs and $2041.1 \pm 0.7$ µs. Thus, the first time constant is similar in $H_2O$ and $D_2O$, but a significant isotope effect is observed for the two others ($k_H/k_D = 2.45$ and 2.47, respectively) which can be assigned to proton transfer steps. Decay associated spectra (Fig. 2d) show that the 5.57 µs time constant in $H_2O$ solution is mainly characterized by a growth of the positive band at 460 nm and a decay at 390 nm. The 36.1 µs time constant has some positive and negative contributions characteristic of the band shift observed, while the 825 µs time constant is mainly characterized by the respective decay and growth of the 390-nm and 480-nm bands.

Nanosecond transient absorption data were also recorded from a suspension of rsEGFP2 microcrystals (in 100 mM HEPES pH 8, 2.5 M ammonium sulfate), using a modified flash-photolysis setup (see Methods). Transient difference spectra similar to those measured in solution were obtained, with positive bands first growing at 460 nm and then at 480 nm (Supplementary Fig. 6). A global fit analysis of kinetic traces for all wavelengths yield time constants of $4.75 \pm 0.13$ µs, $42.9 \pm 0.8$ µs and $295 \pm 2$ µs, i.e. significantly smaller than of rsEGFP2 in $H_2O$ solution (50 mM HEPES pH 8, 50 mM NaCl). To test whether these results are due to the differences in buffer composition or in protein state, measurements were made with a solution of rsEGFP2 in buffer containing ammonium sulfate at a concentration too low to produce crystals or cause protein precipitation. In this buffer (50 mM HEPES, 50 mM NaCl, pH 8, 1.25 M ammonium sulfate), time constants of $4.23 \pm 0.02$ µs, $40.4 \pm 0.2$ µs and $245.2 \pm 0.2$ µs were found (Supplementary Fig. 7), which are similar to those for microcrystals. This suggests that ammonium sulfate accelerates proton transfer in rsEGFP2.

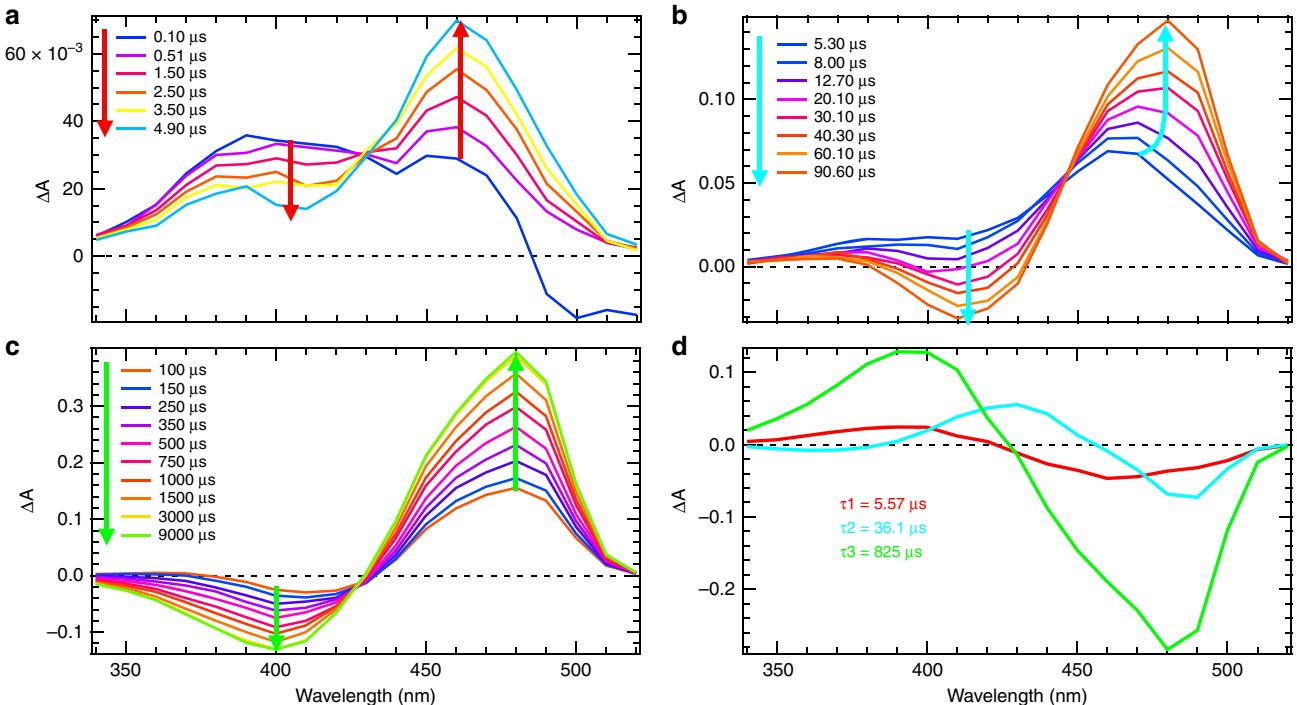

**Fig. 2 Transient UV–Visible spectroscopy in H$_2$O solution.** Time-resolved difference absorption spectra of rsEGFP2 in H$_2$O solution (50 mM HEPES pH 8, 50 mM NaCl) recorded after a 410 nm nanosecond excitation of the *trans*-protonated off-state in the time windows from 100 ns to 9 ms (**a–c**). The spectrum without laser excitation was subtracted to calculate the difference spectra. The colored arrows (red in (**a**), cyan in (**b**) and light green in (**c**)) correspond to the three time constants (5.57, 36.1 and 825 μs, respectively) identified by a global fit analysis of kinetic traces for all wavelengths. **d** Decay associated spectra obtained by fitting the kinetic traces in panels (**a–c**) for all wavelengths with a weighted sum of three exponential functions.

**rsEGFP2 structure solved by TR-SFX 10 ns after photoexcitation.** In order to structurally characterize intermediates along the off-to-on photoswitching reaction coordinate of rsEGFP2, TR-SFX was carried out at the Spring-8 Angstrom Compact Laser (SACLA[39]) XFEL, according to an optical pump–X-ray probe scheme (Experiment No. 2015A8031, July 2015). A pump–probe delay of 10 ns was specifically chosen in order to increase chances to characterize a ground-state intermediate featuring a *cis*-protonated chromophore. Indeed, the 10-ns delay is much shorter than the μs–ms timescale on which time-resolved UV–visible transient absorption spectroscopy identified chromophore deprotonation to occur (see above), while much longer than the ps timescale on which a twisted-chromophore isomerization intermediate has been characterized in the excited state[27]. Prior to injection, rsEGFP2 microcrystals were photoswitched from their (resting) on-state to their off-state (half-life of 100 min[40]) by illumination with 488-nm light, using a dedicated pre-illumination device[40]. Two interleaved datasets were collected, with (laser_on_Δ10 ns) and without (laser_off) activation by the 400-nm pump laser (peak power density 1.25 GW/cm$^2$). From the laser_off dataset, a laser_off structure was determined at 1.60 Å resolution. The initial model consisted of 90% off-state and 10% on-state (see Method section for details), with the chromophore being in the *trans* and the *cis* conformation, respectively (Fig. 3a). The reason for including the on-state model was because of absorption spectroscopy that indicated a residual amount of ~10% of the molecules remained in the on-state after pre-illumination[40]. Moreover, in the resulting mF$_{obs}^{laser\_off}$–DF$_{calc}^{laser\_off}$ map, negative peaks on the *trans* chromophore and on the side chains of Tyr146 and His149 were observed, suggesting that the off-state is even less than 90% occupied (Fig. 3a). Additionally, a positive peak at 4.2 σ was observed halfway between the *trans* and *cis* conformers of the

chromophore, suggesting the presence of a third, hitherto unobserved conformation (Fig. 3a). An ensemble refinement against the laser_off dataset was carried out starting from the *trans* chromophore of the off-state and also suggested the presence of a third chromophore conformation (oval contour in Fig. 3b). Ensemble refinement identified this conformation as a *trans* isomer (called *trans2* hereafter). Thus, the chromophore was modeled in a triple conformation, i.e. *trans1*, *trans2*, and *cis* and refined against the laser_off dataset, yielding occupancies of 65, 25 and 10%, respectively (Fig. 3c). Ensemble refinement also suggested a third conformation of the His149 side chain (His149-supp, cf. circle in Fig. 3b), in addition to the ones characteristic of the on- (His149-on) and the off- (His149-off) states (Fig. 3b, c), respectively. Likewise, three conformations of His149 (His149-off, His149-supp, His149-on) were included in the laser_off model at 65, 25, and 10% occupancy, respectively. Inclusion of these new chromophore and His149 conformers cleared all major peaks in the F$_{obs}^{laser\_off}$–F$_{calc}^{laser\_off}$ map (Fig. 3d). Three alternate conformations were also included for residues at positions 146-148 and 150-152 of β-strand-7, which is known to adopt slightly different conformations in the on- and the off-states at room-temperature[27], residues at the N- and C-terminus of the chromophore (residues 65 and 69), and for Thr204 and Glu223 (see details in the Methods section).

Following pump-laser irradiation, some protein molecules in the crystal are excited and change structure whereas others are not and thus remain in the laser_off structure. The structural features that occur within 10 ns after photoexcitation can be disentangled from the resting laser_off structure in a q-weighted[41] difference Fourier map. Such a map was calculated at 1.85 Å resolution, using observed structure factor amplitudes of the laser_on_Δ10 ns and laser_off datasets (F$_{obs}^{laser\_on\_\Delta10ns}$–F$_{obs}^{laser\_off}$) and phases calculated from the

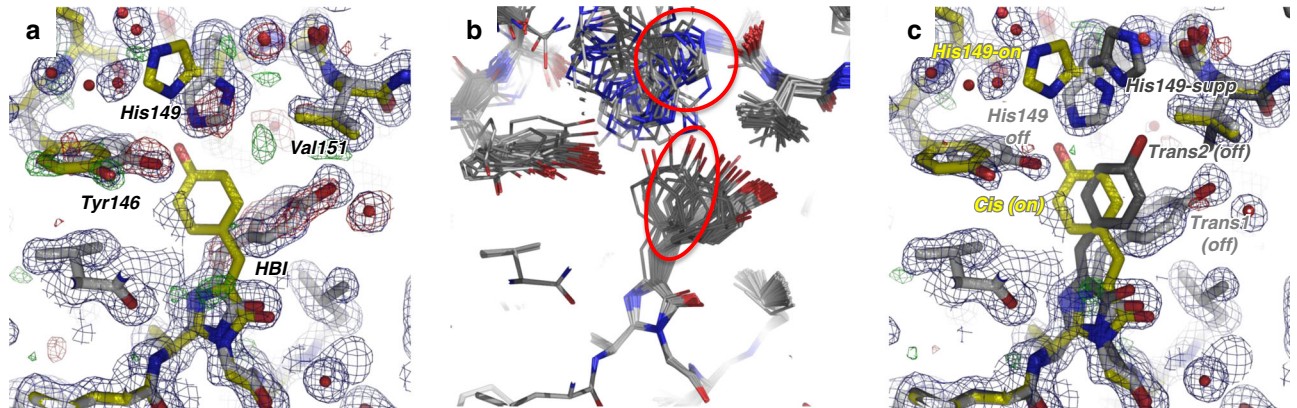

**Fig. 3 Off-state structure.** Chromophore (HBI) and its neighboring residues in the rsEGFP2 laser_off structure determined by SFX without pump-laser activation are shown. **a** The initial laser-off model consists of a mixture of the off-state conformer (light gray carbon trace) and the residual on-state conformer (yellow carbon trace) with occupancies of 90% and 10%, respectively. $2F_{obs}^{laser\_off} - F_{calc}^{laser\_off}$ (blue) and $F_{obs}^{laser\_off} - F_{calc}^{laser\_off}$ (green/red) maps at 1.6 Å resolution are displayed at 1σ and ±3σ, respectively. A positive $F_{obs}^{laser\_off} - F_{calc}^{laser\_off}$ peak between the *trans* (gray) and the *cis* (yellow) chromophores suggests the presence of an additional conformer. **b** Result of ensemble refinements against the laser_off dataset starting for the off-state model (chromophore 100% *trans1*). A third chromophore conformation and a third rotamer of His149 are revealed. (**c**) The final laser-off model features triple conformations of His149 and of the chromophore, i.e. His149-off and *trans1* (light gray), His149-on and *cis* (yellow) and the additional His149-supp and *trans2* (dark gray) conformations, at 65%, 10% and 25% occupancy, respectively. $2F_{obs}^{laser\_off} - F_{calc}^{laser\_off}$ (blue) and $F_{obs}^{laser\_off} - F_{calc}^{laser\_off}$ (green/red) maps calculated from the laser-off dataset at 1.6 Å resolution are displayed at 1σ and ±3σ, respectively.

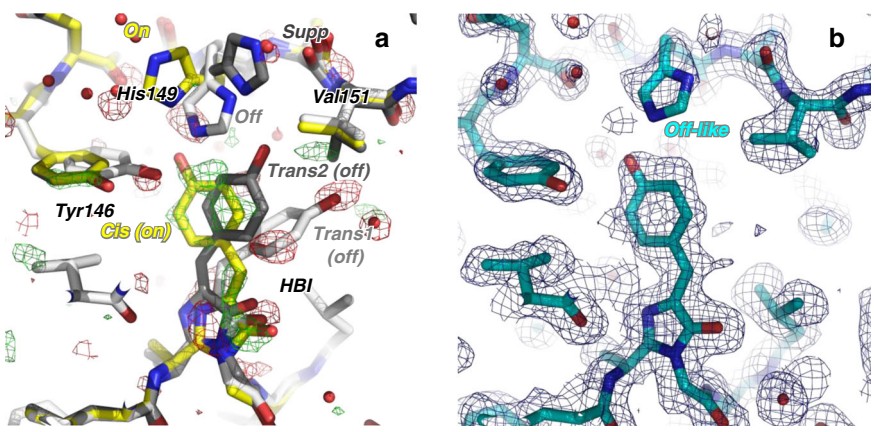

**Fig. 4 10-ns intermediate-state structure. a** Q-weighted difference electron density map ($F_{obs}^{laser\_on\_\Delta10ns} - F_{obs}^{laser\_off}$), determined from SFX data with and without pump-laser activation, is contoured at +3σ (green) and −3σ (red) and overlaid onto the model determined from the laser_off dataset. **b** Model of the laser_on_Δ10 ns intermediate structure (cyan) determined by difference refinement at 1.85 Å resolution. $2F_{extrapolated}^{laser\_on\_\Delta10ns} - F_{calc}$ (blue, 1σ) and $F_{extrapolated}^{laser\_on\_\Delta10ns} - F_{calc}$ maps (green/red, ±3σ, respectively) are shown.

laser_off model (Fig. 4a). The strongest features are located at the chromophore and its direct environment. Negative (down to −5.7 σ) and positive (up to 4.5 σ) peaks are observed at the positions of the *trans1* and the *cis* chromophore, respectively, in particular on their phenol group and methylene bridge. We do not observe a negative peak on the *trans2* chromophore. Such a negative peak might have vanished by overlay of a nearby positive peak on the *cis* chromophore (Fig. 4a). Additionally, negative peaks are present on the off-state conformers of Tyr146 (−4.2 σ) and His149 (−4.8 σ on His149-off), and a positive one on the on-state conformers of Tyr146 (3.5 σ). The off-state has thus been depleted and an intermediate-state built up with a chromophore conformation similar to the one observed in the on-state.

A structural model of the rsEGFP2 intermediate structure 10 ns after photoexcitation (laser_on_Δ10 ns intermediate structure) was determined by difference refinement[42,43] using extrapolated structure factor amplitudes ($F_{extrapolated}^{laser\_on\_\Delta10ns}$, see eq. 1 in Methods section)[44]. An occupancy of ~50% was estimated for the intermediate state in the laser_on_Δ10 ns dataset

(Supplementary Fig. 8, see details in the Methods section). The $2F_{extrapolated}^{laser\_on\_\Delta10ns} - F_{calc}$ map (Fig. 4b) features continuous and well-defined electron density in the intermediate state for the entire chromophore and for all neighboring residues, including His149. At 10 ns, the chromophore has transitioned to a *cis* isomer. The main differences between the off-state and the laser_on_Δ10 ns intermediate structures, besides chromophore isomerisation, are structural changes in the side chains of Tyr146 and His149 (Fig. 5). The former is found in its on-state conformation, whereas the latter remains in an off-like conformation. Figure 6 displays an overlay of the laser_on_Δ10 ns intermediate structure and the on-state structure (PDB entry code 5O89[27]) and shows that the main difference between these are the conformations of His149 and Glu223. In the on-state, the δ nitrogen atom (ND1) of His149 H-bonds to the deprotonated chromophore phenol oxygen (Fig. 6c), whereas in the laser-on-Δ10 ns intermediate structure (Fig. 6b) it is the ε (NE2) nitrogen atom of His149 that H-bonds to the phenol oxygen of the chromophore (distance 2.4 Å). Assuming that His149 is singly

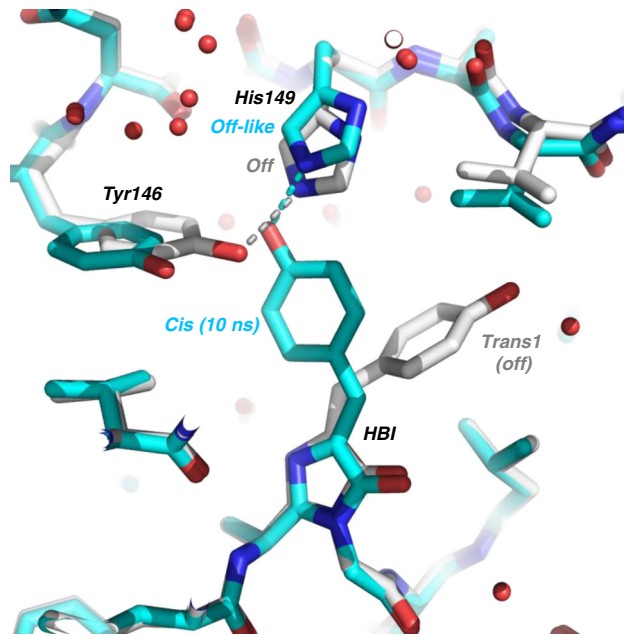

**Fig. 5 Overlay of 10-ns intermediate-state and off-state structures.** The chromophore regions in the 10-ns intermediate structure (cyan) and the laser-off model (gray) are shown. For the sake of clarity, only the major conformer (65% occupancy) of the laser-off model is shown. Note that His149 is hydrogen bonded to Tyr146 in the laser-off model and to the chromophore in the 10-ns intermediate structure.

protonated and that its protonation state does not change between the laser_on_Δ10 ns intermediate and the on-state structures, i.e. that His149NE2 is unprotonated and His149ND1 protonated, the chromophore phenol would be an obligate donor for the H-bond to His149NE2 in the laser-on-Δ10 ns intermediate structure. Thus, the His149 conformer observed 10 ns after excitation by the pump laser are compatible with a protonated *cis* isomer of the chromophore. As to Glu223, one of its carboxyl oxygens is in H-bonding distance to N2 of the chromophore (3.1 Å) in the laser_on_Δ10 ns intermediate (Fig. 6e) but not in the on-state structure (Fig. 6f), and is in H-bonding distance to Ser206OG in the on-state structure (Fig. 6f) but not in the laser_on_Δ10 ns intermediate structure (Fig. 6e).

## Discussion

The mechanism of off-to-on photoswitching in rsEGFP2 involves *trans*-to-*cis* isomerization and deprotonation of the chromophore, and it was here investigated by time-resolved pump–probe UV–visible spectroscopy and TR-SFX. Spectroscopy in solution provides evidence for the existence of four intermediate states decaying with time constants of 87 ps, 5.57, 36.1 and 825 µs, respectively. The latter two time constants are markedly affected when experiments are carried out in $D_2O$ instead of $H_2O$ solution, changes compatible with a proton transfer step. The data thus show that chromophore deprotonation takes place after excited-state decay in the ground state, with time constants of 36.1 and 825 µs. The crystal structure obtained 10 ns after phototriggering features a *cis* chromophore that is still in the protonated state, as inferred from the spectroscopic data and from

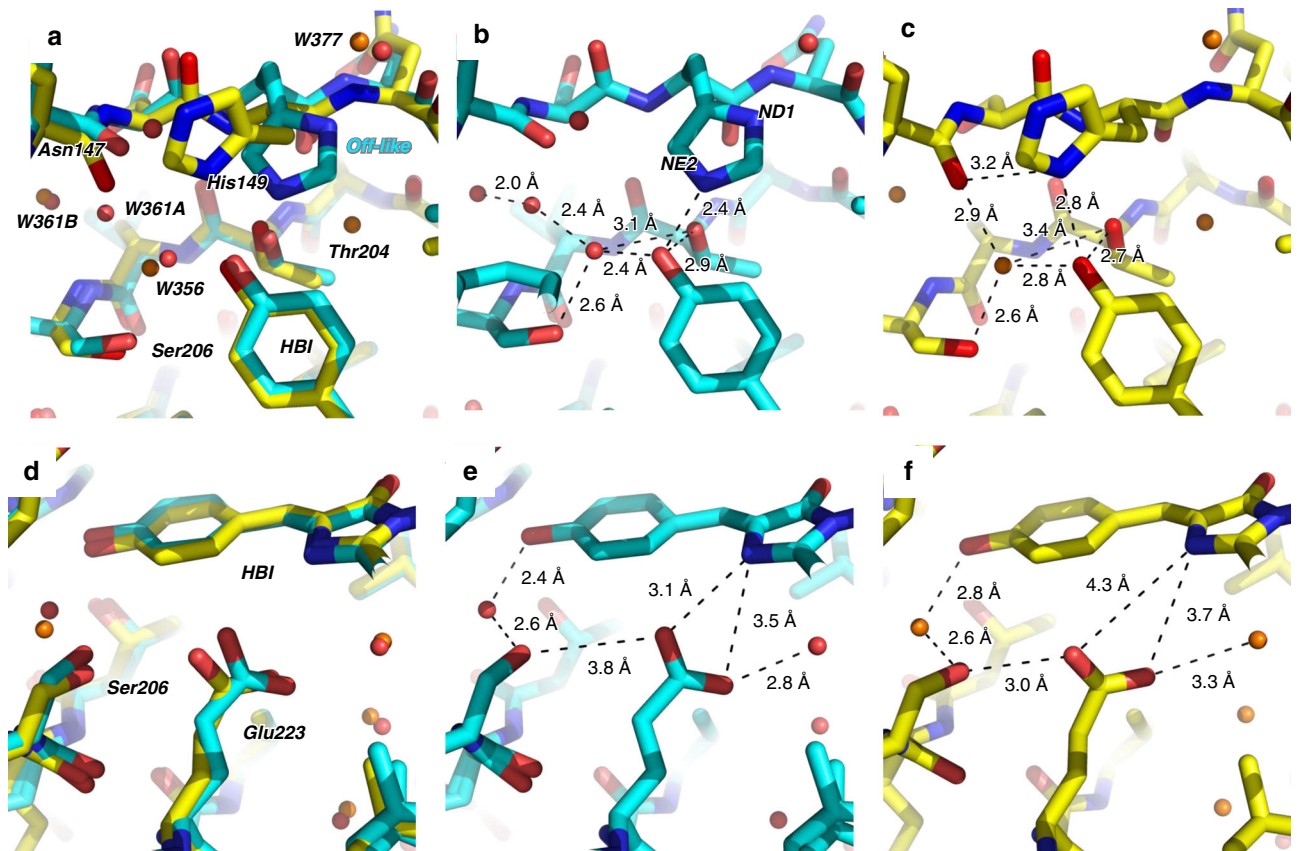

**Fig. 6 Overlay of 10-ns intermediate-state and on-state structures.** Overlay (**a**, **d**) of the chromophore region in the 10-ns intermediate structure (cyan, **b**, **e**) and the on-state structure (yellow, **c**, **f**) determined from SFX at room temperature (PDB entry code 5O89[27]). The focus is on the phenol moiety of the chromophore, His149 and water W356 (**a–c**) and on the entire chromophore, Ser206 and Glu223 (**d–f**). Key distances are indicated.

analysis of the hydrogen-bonding network at the chromophore phenol group. In our earlier TR-SFX study[27], chromophore twisting was observed in the excited state 1 ps after photoexciting the off-state. Also, a low-occupancy population of a *cis* isomer was found 3 ps after photoexcitation that indicated chromophore isomerisation occurs in the excited state. Due to insufficient data quality, however, no structural model could be built of the 3-ps intermediate state[27]. The present spectroscopic and structural data unequivocally shows that the excited-state chromophore isomerisation in rsEGFP2[27] is followed by proton transfer in the ground state.

The TR-SFX structure determined from data collected 10 ns after triggering differs from the structure of the starting off-state (Fig. 5) and of the stable *cis* anionic on-state (Fig. 6), respectively, and corresponds to that of an intermediate state (laser_on_Δ10 ns). Spectroscopy (see above) provides evidence that the chromophore is still protonated at 10 ns (Supplementary Fig. 2 and Fig. 2) so that the 10-ns intermediate corresponds to a *cis* protonated photoproduct. In the 10-ns intermediate structure, the side chain of His149 remains in an off-like conformation while the chromophore has isomerized (Figs. 4b and 6b). His149 might be more disordered in the intermediate than in the off-state structure, since there is a negative but no positive peak on that residue in the $F_{obs}^{laser\_on\_\Delta10ns} - F_{obs}^{laser\_off}$ map. The transition to the final on-state conformation of His149 is thus completed on a timescale longer than 10 ns. Notably, the water molecule which is being hydrogen bonded to the chromophore phenolate in the on-state (W356 in Fig. 6a), and absent in the off-state (Fig. 5), is present at 10 ns (Fig. 6b). Interestingly, Thr204 adopts the same rotamer in the 10-ns intermediate and the on-state structure (Fig. 6a), whereas it is in different rotamers in 1-ps intermediate and in the off-state structure[27]. In the 10-ns intermediate, the chromophore phenol group engages in three H-bonds, i.e. His149NE2, W356 and Thr204OG1 (Fig. 6b).

The 50% occupancy of the laser_on_Δ10 ns intermediate exceeds the primary quantum yield of the off-to-on photoswitching reaction (0.33[18]). Indeed, the pump-laser parameters chosen resulted in nominally 17 absorbed photons per chromophore (cf. Methods) so that those molecules that after excitation return to a hot *trans* ground state within a few ps[27] and then further evolve to the cold initial *trans* ground state can be excited again during the 100-ps pump pulse. Additionally, intermediates formed within 100 ps can also absorb and either drive the reaction forward or generate off-pathway intermediates. The time-resolved difference absorption spectra recorded at 40 and 100 ps display values close to zero at 400 nm (Supplementary Fig. 1a), indicating that the signals from ground-state depletion and an arising intermediate state absorbing at 400 nm compensate. Therefore, we cannot exclude that the laser_on_Δ10 ns intermediate has partially evolved from an intermediate absorbing at 400 nm during the 100 ps pump pulse.

During the transition from the laser_on_Δ10 ns intermediate to the stable *cis* anionic on-state His149 moves from the off-like conformation (Fig. 7b) to its on conformation (Fig. 7d). This move could correspond to the process with a 5.57 µs time constant determined by spectroscopy. Alternatively, the 5.57 µs process could correspond to structural rearrangements elsewhere in the protein and His149 would move to the on position concomitantly with chromophore deprotonation. A TR-SFX experiment with a pump–probe delay of 20 µs would provide the unknown structure (Fig. 7c) prior to chromophore deprotonation, therefore allowing to decide between the two alternative scenarios. Changes in the H-bonding network around the chromophore phenol group between the laser_on_Δ10 ns intermediate and the *cis* anionic on-state structure (Fig. 6) are strikingly similar to the ones between the protonated A form and

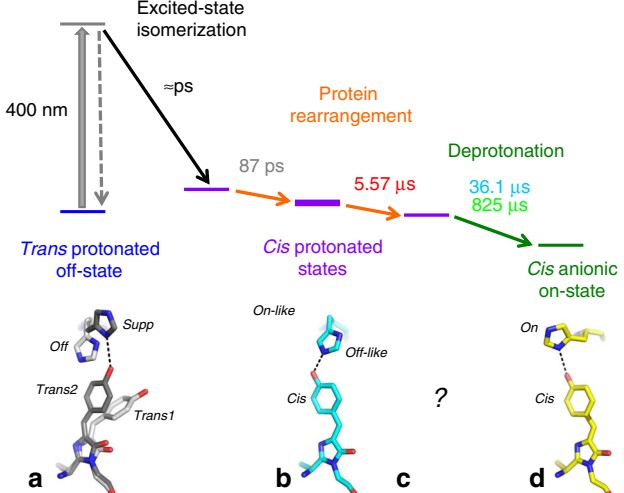

**Fig. 7 Model for the rsEGFP2 off-to-on photoswitching process.** The bold purple bar represents the laser_on_Δ10 ns intermediate structure determined by TR-SFX. Time constants correspond to those determined by femtosecond (87 ps) and nanosecond-resolved pump–probe UV–visible absorption spectroscopy (5.57, 36.1 and 825 µs) in $H_2O$ solution (50 mM HEPES pH 8, 50 mM NaCl). All three *cis* protonated states (purple bars) are ground-state intermediates, interconversion of which involves rearrangements of the protein matrix (orange arrows). Corresponding structural models for the chromophore and His149 as determined by X-ray crystallography are shown in panels **a**, **b** (this work) and **d**[27]. *Trans1* and His149-off (light gray) and *trans2* and His149-supp (dark gray) of the trans-protonated off-state (a, Fig. 3c) transited to *cis* and His149-off-like at 10 ns (**b**, Fig. 4b, Fig. 6b, e). His149-off-like has moved to His149-on in the *cis* anionic *on* state (**d**, Fig. 6c, f). The structure whose intermediate forms in 5.57 µs remains elusive (**c**).

the anionic B form of wild-type GFP[45]. An exception is the H-bond between the chromophore phenol and Thr204OG1 present in the laser_on_Δ10 ns intermediate structure but absent in the protonated A form of wild-type GFP (Fig. 2a in[45]). The laser_on_Δ10 ns intermediate structure allows proposing three different pathways taken by the proton upon chromophore deprotonation. On the first pathway, the proton is transferred from the chromophore via Thr204OG1 and His149O to the solvent outside the protein as suggested for GFP[46]. A proton pathway between the solvent and the chromophore has also been suggested to exist in the positive RSFP Padron[47,48]. A second pathway proceeds via His149 and a chain of three water molecules (W377, W158, W144; only W377 is seen in Fig. 6a) connected to His149ND1 to the solvent. A third pathway guides the proton via a Grotthuss mechanism involving W356 and W361 (Fig. 6a, b) to the solvent.

The laser_off structure, determined from SFX data without pump-laser excitation, features the chromophore in two *trans* conformations, one of which (*trans2*, 25% occupancy) was not previously observed in the off-state of rsEGFP2. The third His149 conformation (His149-supp), included in the laser_off structure in addition to the ones in the on (His149-on) and the off (His149-off) states (Fig. 3c) might correspond to the *trans2* conformation of the chromophore to which it would be H-bonded (distance of 2.7 Å between phenol oxygen and ND1). Ensemble refinements against published off-state datasets collected by room-temperature SFX[27] and cryo-crystallography at a synchrotron[18] show no indication of *trans2* and His149-supp conformations (Supplementary Fig. 9). We do not yet have an explanation for this structural off-state heterogeneity. Evidence for the existence of a *trans2* chromophore also comes from

cryo-crystallographic synchrotron experiments, in which illumination at 488 nm of an rsEGFP2 variant with a substituted phenolate ring yielded *trans2* when crystals with a contracted unit cell were examined, whereas the *trans1* conformation is populated when crystals with a larger unit cell were used[49]. The authors suggest the tighter packing of proteins in the contracted crystal reduces the flexibility of the chromophore pocket, thereby limiting the volume available for the isomerizing chromophore and favoring the volume-conserving *cis*-to-*trans2* transition, rather than the volume-consuming *cis*-to-*trans1* transition. Coincidentally, different illumination conditions were used for the two datasets. We note that a chromophore conformation similar to *trans2* has been observed in rsFolder[18], a reversibly switchable fluorescent protein designed for efficient expression in oxidizing cellular environments.

Based on the new results described here, we can now propose a model for the rsEGFP2 off-to-*on* photoswitching process (Fig. 7). Excitation at 400 nm of the trans-protonated chromophore in the off-state triggers excited-state isomerization to a *cis* chromophore within a few picoseconds[27]. After excited-state decay, there is an evolution in the ground-state with a 87 ps time constant that we assign to a protein reorganization to accommodate the *cis* chromophore. The resulting intermediate state with a *cis* protonated chromophore remains stable up to 100 ns (Supplementary Fig. 2) and was characterized by TR-SFX at 10 ns (purple bold bar in Fig. 7). Transient absorption spectroscopy in $H_2O$ solution evidences a further ground-state process with a time constant of about 5.57 μs (no isotope effect) that might correspond to a protein rearrangement during which the off-like conformation of His149 transitions to the on conformation (Fig. 7). Alternatively, the conformational change in His149 could accompany chromophore deprotonation that occurs with characteristic time constants of 36.1 and 825 μs (isotope effect) to yield the final *cis* anionic on-state.

Together, the present and the preceding ultrafast studies[27] clarify the order of off-to-on photoswitching events in rsEGFP2 (hydrozoan origin) and provide evidence for chromophore isomerization and deprotonation being excited-state and ground-state processes, respectively. If the photoswitching mechanism of RSFPs from Anthozoa is identical to the one from Hydrozoa, our study would disagree with the proposed ground-state isomerization in Dronpa[14,15] and provide unambiguous support to the proposed excited-state isomerisation and ground-state deprotonation[11–13]. TR-SFX on Dronpa and TR-IR spectroscopy on rsEGFP2 would shed light on differences and similarities in the photoswitching mechanisms of RSFPs from Anthozoa and Hydrozoa.

## Methods

**Protein expression, purification, and crystallization**. rsEGFP2[16] fused to an N-terminal polyhistidine tag was expressed in *E. coli* BL21 cells and purified by Ni-NTA affinity and size exclusion chromatographies[18]. For TR-SFX, rsEGFP2 microcrystals ($3 \times 3 \times 3$ μm³) were generated by seeding a solution with final protein, precipitant, salt and buffer concentrations of ~20 mg/ml, ~2 M ammonium sulfate, 20 mM NaCl, 120 mM HEPES pH 8, respectively. Sedimented microcrystals were resuspended in 100 mM HEPES pH 8, 2.5 M ammonium sulfate prior to injection into the XFEL beam. A suspension containing 2–8% (v/v) microcrystals was filtered through a 20-μm stainless steel filter using a sample loop and a manually driven syringe.

**$H_2O$/$D_2O$ exchange of rsEGFP2 in solution**. $H_2O$/$D_2O$ exchange for nanosecond transient absorption spectroscopy was performed by several cycles of dilution/concentration. 450 μl of 50 mM HEPES pD 8, 50 mM NaCl were added to a tube containing 50 μl of rsEGFP2 at 20 mg/ml in 50 mM HEPES pH 8, 50 mM NaCl. Then the protein solution was concentrated to a final volume of 100 μl using 10 kDa concentrators (Amicon 1 ml Centrifugal Filters 10 kDa Millipore). The dilution/concentration cycle was repeated four times, so that the total dilution factor was at least 1000.

**Time-resolved UV–visible absorption spectroscopy**. Femtosecond transient absorption spectroscopy from 40 ps to 2 ns was carried out on rsEGFP2 (12.5 mg/ml) in a $H_2O$ solution containing 50 mM HEPES pH 8, 50 mM NaCl according to the protocol reported earlier[27] (Supplementary Fig. 1).

Nanosecond transient absorption spectroscopy experiments were carried out from 10 ns to 10 ms (Fig. 2, Supplementary Figs. 2, 5, 7) using a conventional flash-photolysis setup (90° geometry between pump and probe beams, Supplementary Fig. 10)[50] on three rsEGFP2 solutions (46 μM): (i) 50 mM HEPES pH 8, 50 mM NaCl in $H_2O$, (ii) 50 mM HEPES pD 8, 50 mM NaCl in $D_2O$, (iii) 50 mM HEPES pH 8, 50 mM NaCl, 1.25 M $(NH_4)_2SO_4$ in $H_2O$. Initially, the solutions (2 ml) were illuminated for 3 min at 488 nm (Cobolt 06-MLD 488; 200 mW, 2 mm $1/e^2$ waist diameter) to photoswitch 90% of the proteins to the *trans*-protonated off-state (absorbance of about 1 at 410 nm for a 1 cm thickness). Pump beam (410 nm, 8 ns, 1.6 mJ, beam size approximately 1 mm × 10 mm) was focused using a cylindrical lens onto a spectroscopic cell (Hellma, chamber thicknesses 1 mm) containing the protein solution (Supplementary Fig. 10a). The time traces of the variation in absorbance were recorded from 350 to 520 nm in 10-nm steps (3.5 nm spectrometer bandpass) to reconstruct the transient difference absorption spectra at different time delays after the pump-pulse excitation. The recorded time traces were obtained by averaging a sequence of eight pump–probe experiments (laser energy fluctuation is below 0.1 mJ pulse-to-pulse and is not corrected). There was a time lapse of 30 s between two consecutive pump-pulse excitations. To photoswitch the proteins in the stationary solution back to the off-state after each 410 nm pump pulse the spectroscopic cell containing the stationary protein solution was illuminated continuously by 490-nm light (collimated Thorlabs LED M490L4, 200 mW, beam diameter 9 mm; Supplementary Fig. 10a). Time traces of the variation in absorbance can be analyzed at and above 10 ns, given the 8-ns pump pulse length. A transient difference absorption spectrum at one time delay (t) is obtained by averaging 10 time delay points (from t−5 points to t + 5 points) of the time traces. Three time delay windows were explored from 10 ns to 5 μs (500 ps/point), from 5 μs to 100 μs (10 ns/point) and from 100 μs to 10 ms (1 μs/point).

For spectroscopy experiments on microcrystal suspensions (Supplementary Fig. 6), rsEGFP2 microcrystals ($3 \times 3 \times 3$ μm³) were suspended in 100 mM HEPES pH 8, 2.5 M ammonium sulfate. Suspensions were initially irradiated for 30 min at 488 nm (Cobolt 06-MLD 488; 200 mW, 2 mm $1/e^2$ waist diameter) to achieve off-switching. For the subsequent ns–ms transient absorption experiment, the microcrystal suspension was placed in a spectroscopic cell (Hellma, chamber thickness 1 mm, Supplementary Fig. 10b). The conventional flash-photolysis setup was modified (15° between pump and probe beams, Supplementary Fig. 10b) to minimize light scattering. The pump laser (410 nm, 8 ns, 5 mJ, beam diameter approximately 2 mm) was focused using a spherical lens onto the cell containing the microcrystal suspension. The probe light-source was not modified from conventional experiments. The diameter of both pump and probe beams was 2 mm through the spectroscopic cell. Following single-shot excitation, the time traces of the variation in absorbance were recorded from 350 to 520 nm in 10-nm steps (9.4 nm spectrometer bandpass) to reconstruct the transient difference absorption spectra at different time delays after the pump-pulse excitation. Between each single-shot excitation the cell was moved manually to probe a fresh area. Time traces of the variation in absorbance start to be meaningful at 500 ns. Time windows explored were from 500 ns to 100 μs (10 ns/point) and from 100 μs to 2 ms (200 ns/point).

For the analysis of transient absorption spectra, a model based on a weighted sum of three exponential decays was chosen to fit the kinetic traces for all wavelengths (global fit analysis). This was done with Igor Pro and a custom-made analysis routine in python 3 using lmfit package[51]. The quality of the fit was checked by analyzing the residuals (no structure, Supplementary Fig. 3) and the Levenberg-Marquardt algorithm was used to minimize least-squares error surface ($\chi^2 < 10^{-5}$). The reported values of the standard deviations of the parameters correspond to the diagonal elements of the covariance matrix, which is the inverse of the so-called curvature matrix taken at the minimum of the error. By contrast to linear fitting, this matrix is not constant but a function of the fitting parameters for nonlinear modeling. Still, information on standard deviations can be derived but should, however, be interpreted with caution, keeping in mind the assumption that the set of parameters estimated is true. A more reliable estimation of the uncertainty would require replicated estimations of the nonlinear parameters on different datasets. These replicated datasets can be generated with bootstrap. Fitting these datasets provides different sets of parameters, whose means and standard deviations can be estimated from their distributions. The model was here fitted on 1000 replicated datasets constructed using the best parameters of the fit and adding replicates of noise (Supplementary Fig. 4). These replicates were obtained from the residuals (difference between the original and reconstructed data) randomly shuffling 25% of the values. Figures were made using Igor Pro and Searborn and matplotlib python 3 package.

**SFX data collection**. A suspension of rsEGFP2 microcrystals (2–8% (v/v)) was transferred into a stainless steel syringe that was mounted on an anti-settling device[52] and whose temperature was maintained at 20 °C with an adjustable Peltier element. The microcrystal suspension was injected with a gas dynamic virtual nozzle (GDVN)[53] injector into the helium-filled Diverse Application Platform for hard X-ray Diffraction in SACLA (DAPHNIS)[54] chamber on beamline BL3-EH4c

at SACLA[39] (hutch temperature ca. 27 °C). The GDVN carried a sample capillary of 100 μm inner diameter and operated at flow rates of 30–40 μl/min. Both parameters, as well as the low microcrystal concentration, were critical to avoid microcrystal aggregation and thus nozzle-clogging that occurred at lower flow rates and smaller nozzle diameters. The resulting jet had a diameter of 3–5 μm at a focusing helium pressure of 15 psi.

On their way from the sample syringe to the injector, rsEGFP2 microcrystals were photoswitched from their on-state (the resting state) to the off-state (see Fig. 1) by irradiation with continuous 488-nm laser-light (cw, 200 mW) within a pre-illumination device[40]. The analysis of absorption spectra (Fig. 6 in[40]) obtained from microcrystals collected at the exit of the device (200 mW) and without pre-illumination indicated that approximately 90% of microcrystalline rsEGFP2 were switched off. The transit time from the pre-illumination device to the interaction zone was less than, or equal to, 1 min. This time interval is short compared to the off-to-on thermal recovery of microcrystalline rsEGFP2 (100 min[40]) so that 90% of the proteins were in their off-state when reaching the interaction zone.

SFX data were collected at 30 Hz using an X-ray beam (nominal photon energy 10 keV, pulse length ≤ 10 fs, 300 μJ per pulse at the sample position, $2 \times 10^{11}$ photons/pulse) focused to 1.4 μm (horizontal) × 1.6 μm (vertical) (FWHM). Diffraction data were recorded using an octal-MPCCD detector with eight sensor modules[55] positioned 52 mm away from the sample. Online monitoring of diffraction data, such as determination of hit-rate and estimation of the fraction of multiple hits, was done with CASS[56].

TR-SFX data were collected using an optical pump–X-ray probe scheme with a pump–probe delay of 10 ns. The off-to-on transition was triggered by a pump-laser pulse (400 nm, 100 ps pulse length, 180 μm × 180 μm focal spot (FWHM), 46 μJ per pulse, leading to a peak power density of 1.25 GW/cm² and corresponding nominally to 17 absorbed photons per chromophore) generated by a Ti:sapphire laser, aligned perpendicularly to both the liquid jet and the X-ray beam. A sequence of interleaved laser_off and laser_on diffraction images was collected at 30 Hz.

**SFX data processing.** A total of about 609,000 frames were collected for the laser_on_Δ10 ns and the laser_off datasets, respectively. NanoPeakCell[57] was used to perform offline hit-finding and sort frames with and without prior pump-laser excitation into laser_on_Δ10 ns (13300 hits) and laser_off (12487 hits) datasets, respectively. CrystFEL 0.6.2 was used for further data processing (i.e. indexing and integration), resulting in 9781 and 9997 indexed frames for the laser_on_Δ10 ns and laser_off datasets, respectively. The "ring-nocen" method was used for intensity integration. The sample-detector distance was refined until the distribution of unit-cell parameters was Gaussian[58]. Merging with the partialator module of CrystFEL 0.6.2 was optimal when combining scaling and partiality refinement with the push-res option set to zero, as judged by $R_{split}$, CC* and the Wilson plot. Data collection statistics are given in Supplementary Table 1.

**Structure solution and refinement.** The structure corresponding to the laser_off dataset was phased by molecular replacement using *Phaser*[59] with the structure of rsEGFP2 in it off-state determined by synchrotron cryo-crystallography (PDB entry 5DTY[18]) as a starting model. Refinement with the *Phenix* suite[60] included positional and isotropic individual B factor refinement in reciprocal space. Model building was carried out using *Coot*[61] and occupancies were set manually. Given that the pre-illumination efficiency was approximately 90%[40], an alternate chromophore conformation corresponding to the *cis* isomer in the on-state was included at 10% occupancy. Accordingly, those residues adopting different conformations in the on- and off-states, i.e. part of β-strand 7 (residues 146 to 152), residues at the N- and C-terminus of the chromophore (residues 65 and 69), and Thr204, were also first modeled in double conformations, (90% off and 10% on). When the remaining 90% were attributed to the *trans* isomer of the off-state, a negative peak in the $F_{obs}^{laser\_off} - F_{calc}^{laser\_off}$ map indicated that the relative occupancy must be lower than 90% and a positive $F_{obs}^{laser\_off} - F_{calc}^{laser\_off}$ peak halfway between the *cis* and the *trans* (called *trans1* hereafter) isomers indicated the presence of a third chromophore conformer (called *trans2* hereafter) occupied at 25% or less (Fig. 3a). Ensemble refinement against the laser_off dataset was carried out with the *Phenix* suite starting from the off-state model (*trans1* chromophore; Fig. 3b). The values of *pTLS*, *tbath* and *tx* were varied, where *pTLS* is the fraction of the molecules included in the TLS fitting procedure, *tbath* is a parameter that controls the X-ray weight and that is coupled to the thermostat temperature and *tx* is the relaxation time used during the simulation. The tested values were 0.6, 0.8, 0.9 and 1.0 for *pTLS*, 2.5, 5 and 10 K for *tbath*, and 0.35, 0.7 and 1.4 ps for *tx*. $R_{free}$ was plotted as a function of the different sets of parameter values using a custom-made python script. The lowest $R_{free}$ was chosen as a criterion to determine the most appropriate set of values of these three parameters. Ensemble refinement indicated the presence of a third chromophore conformation (*trans2*) and a third rotamer of His149 (His149-supp). Consequently, the chromophore and His149 were modeled with three alternate conformations, i.e. *trans1*, *cis*, *trans2*, and His149-off, His149-on, His149-supp, whose occupancies were jointly refined to 65% (alternate conformers A), 10% (alternate conformers B) and 25% (alternate conformers C), respectively (Fig. 3c). Thr204 and residues in β-strand 7 (positions 146-148 and 150-152) and at the N- and C-termini of the chromophore were also modeled in three alternate conformations and included in the occupancy

refinement. As a control, ensemble refinement was carried out against the off-state data collected by cryo-crystallography at a synchrotron[18] starting from the corresponding off-state model (chromophore 100% *trans1*, PDB code 5DTY[18]) and against the off-state data collected by room temperature SFX[27] starting for the corresponding off-state model (PDB code 5O8A) from which the *cis* alternate conformation was omitted so that the chromophore was fully in the *trans1* conformation[27]. The same procedure was applied and the set of parameters yielding the lowest $R_{free}$ values were ptls = 0.8, bath = 5, tx = 2.0 for 5DTY and ptls = 0.8, bath = 5, tx = 2.0 for 5O8A. Neither *trans2* and nor His149-supp, conformations were occupied (Supplementary Fig. 9).

Structural changes 10 ns after pump-laser excitation were qualitatively identified in a q-weighted[41] (qW) difference Fourier electron density map ($F_{obs}^{laser\_on\_\Delta10ns} - F_{obs}^{laser\_off}$), calculated with the laser_off structure as a phase model using CNS[62] (Fig. 4a). For further analysis, extrapolated structure factors ($F_{ext}$) were calculated based on the q-weighted structure factor amplitude differences according to the following formula: $F_{ext}^{laser\_on\_\Delta10ns} = \alpha * q/<q> * (F_{obs}^{laser\_on\_\Delta10ns} - F_{obs}^{laser\_off}) + F_{obs}^{laser\_off}$ (eq. 1), where α is the inverse of the estimated occupancy of the fraction of molecules that changed conformation upon photoexcitation and q is the Bayesian weight retrieved through the q-weighting procedure[41]. To determine α, a procedure reported earlier[27] was applied, i.e. $F_{extrapolated}^{laser\_on\_\Delta10ns}$ were calculated and the ratio of integrated peaks in the $F_{extrapolated}^{laser\_on\_\Delta10ns} - F_{calc}^{laser\_off}$ map and of integrated peaks in the qW $F_{obs}^{laser\_on\_\Delta10ns} - F_{obs}^{laser\_off}$ map was plotted as a function of α (Supplementary Fig. 8). Only peaks in the vicinity of the chromophore, Tyr146, His149, Val151, and Thr204 were integrated. The most appropriate value of α is the one at a maximum is reached, i.e. 2 in the present case (Supplementary Fig. 8). An α of 2 corresponds to an occupancy of 50% of the laser_on_Δ10 ns intermediate structure.

The laser_on_Δ10 ns intermediate structure was refined using the difference refinement procedure, and $2F_{extrapolated}^{laser-on-\Delta10ns} - F_{calc}$ and $F_{extrapolated}^{laser-on-\Delta10ns} - F_{calc}$ electron density maps were used to build the model. The same procedure as for the laser-off model was followed. Refinement statistics are given in Supplementary Table 1. Figures were prepared using PyMOL[63].

## Data availability

Coordinates and structure factors have been deposited in the Protein Data Bank under accession codes 6T39, 6T3A. The source data underlying Supplementary Fig 8 are provided as a Source Data file. Other data are available from the corresponding authors upon reasonable request.

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

## Acknowledgements

The authors are grateful to So Iwata and Rie Tanaka for valuable comments and technical assistance related to the experiments at SACLA and to Hugues Nury for help with sample preparation at SACLA. The XFEL experiments were carried out at BL3-EH4c of SACLA with the approval of the Japan Synchrotron Radiation Research Institute (JASRI; Proposal No. 2015A8031). We warmly thank the SACLA staff for assistance. M.S.l. is grateful to Raffaele Vitale for discussion on bootstrap analysis and Jean-Julien Dubois for technical support in time-resolved spectroscopy experiments and M.W. to Marco Kloos and James Remington for discussions and to Eva Pebay-Peyroula, Jacques Neyton, Gilles Bloch and Renaud Blaise for support. The study was supported by a travel grant from the CEA to J.W., N.C., V.A., E.d.l.M., P.M., J.P.C. and M.W., a grant from the CNRS (PEPS SASLELX) to M.W., an ANR grant to M.W., M.C., M.S.l. (BioXFEL) and a PhD fellowship from Lille University to LMU. We acknowledge support from the Max Planck Society. This work used the Multistep Protein Purification Platform (MP3) of the Grenoble Instruct center (ISBG; UMS 3518 CNRS-CEA-UJF-EMBL) with support from FRISBI (ANR-10-INSB-05-02) and GRAL (ANR-10-LABX-49-01) within the Grenoble Partnership for Structural Biology (PSB). The IBS acknowledges integration into the Interdisciplinary Research Institute of Grenoble (IRIG, CEA) and financial support by the CEA, the CNRS and the UGA. The Chevreul Institute (FR 2638), the Ministère de l'Enseignement Supérieur et de la Recherche, the Région Nord-Pas de Calais and FEDER are acknowledged for financial support. G.N.K. acknowledges support from the E.U. under the program FP7-PEOPLE-2011-ITN NanoMem (project number 317079). J.W. acknowledges support from the CEA through the IRTELIS PhD program.

## Author contributions

M.W., M.S., D.B., J.P.C. and I.S. designed research. S.J. provided rsEGFP2 plasmid. J.W., V.A., V.G., M.T. and F.F. expressed and purified WT protein. J.W., V.G. and I.S. carried out microcrystallization. G.S. and M.B. spectroscopically characterized microcrystals. M.Fe. and M.Fi. carried out simulations. L.U., M.S. and C.R. carried out time-resolved absorption spectroscopy. J.W., G.N.-K., N.C, V.A., T.R.M.B., E.DlM, R.B.D., Y.J., P.M., K.M., K.N., S.O., C.M.R., G.S., R.L.S., T.T., K.T., M.C., L.F., M.S., J.P.C., I.S. and M.W. carried out SFX experiments. R.L.S, R.B.D., G.N.-K. and K.N. performed sample injection. T.T., S.O., and M.C. aligned and controlled the pump laser. Y.J., K.M., K.T., M.Y., L.F. discussed software aspects of experiment, K.M. and L.F. prepared and implemented DAQ and online processing. C.M.R., K.M., L.F., N.C., T.R.M.B. and J.P.C. carried out online SFX data processing. J.W., N.C. and J.P.C. carried out offline SFX data processing. J.W. and J.P.C. carried out structure refinement and analysis, with input from V.A., T.R.M.B., D.B., N.C. I.S. and M.W. J.W. and M.W. wrote the manuscript with contributions and input from D.B., M.S., J.P.C. and I.S. and all authors approved it.

## Competing interests

The authors declare no competing interests.
