## [Peer Review File · Nature Communications]

Reviewers' comments:

Reviewer #1 (Remarks to the Author):

In my opinion, the manuscript by M. Weik and collaborators will be of interest to the community of spectroscopists and structural biophysicists studying light-induced reactions in fluorescent proteins. However I do not think that it will change thinking in this field, since most of the conclusions are not new (see below for more detail). In addition, data presented in the manuscript currently have two major weaknesses that question the author's conclusions (see below). I think this should be solved before the manuscript can be published.

Main results and conclusions:

The manuscript by M. Weik and collaborators is the follow-up of a manuscript published earlier this year by the same authors (Coquelle et al., *Nat. Chem.* 2018, 10, 31). Both manuscripts investigate the mechanism of off→on photoswitching in the reversibly photoswitchable fluorescent protein (RSFP) rsEGFP2 by a combination of XFEL-based time-resolved crystallography and transient absorption spectroscopy. While the first manuscript dealt with primary events in the excited-state (ps timescale), the present one focuses on slower, ground-state reaction steps (from 10 ns to milliseconds).

The main new data are the following:

- The authors solved the structure of rsEGFP2 10 ns after excitation of the off state. The 10-ns structure features a cis chromophore, which indicates that trans→cis isomerization is completed in this time-scale. The protein layout around the chromophore is different from that observed in the final on state, and shows some heterogeneity with 2 conformations of His149 (40 % on-like, 60 % off-like).
- The structure of the initial off state was also reexamined. Surprisingly, it was found to consist of two different trans conformers : trans1, corresponding to the off state structure reported in Coquelle et al. (and in El Khatib et al., *Sci. Rep.* 2016, 6, 18459) and trans2, a hitherto unreported conformer accounting here for 20 % of the population.
- Transient absorption spectra were measured from 100 ns to milliseconds, following excitation of the off state. They reveal 3 kinetic phases : 4.4 μ s, 40 μ s and 919 μ s, with kinetic isotope effects of 1.8 and 2.4, resp., for the two longest components upon hydrogen-deuterium exchange.

The authors conclude from these data 1) that the first ground-state intermediate in rsEGFP2 off→on photoswitching is a cis protonated form corresponding to the structure solved at 10 ns, 2) that the 4.4 μ s component corresponds to the movement of His149 from its off-like to its on-like conformation, and 3) that chromophore deprotonation occurs with a characteristic time of 759 μ s (average of the 40 μ s and 919 μ s components), and is accompanied by a further migration of His149 towards its final on state position.

Novelty and interest:

As far as techniques are concerned, XFEL-based time-resolved crystallography is an exciting new technique that gives access to the light-induced structural dynamics of proteins on time-scales as short as \sim 100 fs. In Coquelle et al., the authors reported structures of rsEGFP2 at 1 ps and 3 ps. In the present manuscript however, they report a structure at 10 ns, a timescale that has already been accessible for \sim 20 years with synchrotron-based time-resolved crystallography (see for instance the 1-ns structure of the photoactive yellow protein in Perman et al., *Science* 1998, 279, 1946). But this is the first nanosecond structure of a RSFP, and therefore I expect that it will be of interest to RSFP specialists.

Concerning the author's conclusions (see list above):

- The cis protonated nature of the first ground-state intermediate in RSFP off→on photoswitching (point 1) was previously established by Yadav et al. (*J. Phys. Chem. B* 2015, 119, 2404) in the case of Dronpa and Dronpa-2, and by Colletier et al. (*J. Phys. Chem. Lett.* 2016, 7, 882) for a third

RSFP, IrisFP. Parallel investigations by femtosecond IR spectroscopy fueled some doubts because they failed to detect the IR signature expected for a cis chromophore (Warren et al., Nat. Commun. 2013, 4, 1461; Lukacs et al., J. Phys. Chem. B 2013, 117, 11954). But it is now established that the lack of a clear signature for the cis chromophore in differential IR spectroscopy is due to the distorted structure of the trans chromophore in the off state (Kaucikas et al., J. Phys. Chem. B 2015, 119, 2350).

- Point 2 (conformational movement of His149) is new and illustrates the interplay between chromophore and protein conformational changes in course of RSFP photoswitching. However, see my comments on timescales below.

- μ s-ms chromophore deprotonation (point 3) was previously reported by Yadav et al. in Dronpa and Dronpa-2 and by Colletier et al. in IrisFP.

With the exception of point 2 which is new and will be of interest to RSFP specialists, the work by M. Weik and collaborators essentially shows that rsEGFP2 has a similar off \rightarrow on photoswitching mechanism as Dronpa, Dronpa-2 and IrisFP. Since rsEGFP2 is a variant of GFP from jellyfish while Dronpa, Dronpa-2 and IrisFP are all variants of fluorescent proteins from corals, this conclusion will also be of interest to specialists.

Major points that question the conclusions:

In my opinion, the two following problems question the manuscript's conclusions and should therefore be solved in priority:

- When reading the manuscript, I was very embarrassed by the inconsistency between the new off state structure and that reported a few months ago by the same authors in Coquelle et al. (and in 2016 in El Khatib et al.). The new, previously unreported trans₂ conformer corresponds to 20 % of the population, this is more than significant. Yet the three structures have similar resolutions (1.7 Å for the present one and that in Coquelle et al., 1.5 Å for El Khatib et al.). Where does this lack of reproducibility come from ? The authors do not give any explanation. In my opinion, the fact that the initial state of the reaction under study is ill-defined raises doubts over the entire work. This point definitely has to be clarified before the manuscript can be published.

- The second major weakness is that there is no temporal match between crystallographic and spectroscopic data, which results in some vagueness in the timescales of rsEGFP2 photoswitching. The structure of the intermediate was taken at 10 ns, while transient absorption was measured from 100 ns to milliseconds. There is also a gap of more than three orders of magnitude with the transient absorption data published in Coquelle et al., which end at 40 ps. The 100-ns spectrum shown in the present manuscript (Figure 2a) is very different from the 40-ps spectrum shown in Coquelle et al. (Supplementary Figure 1c), which suggests that kinetic phases exist for rsEGFP2 in the 40 ps-100 ns time window. These kinetic phases have not been characterized. Yet, the authors claim that the 10-ns structure and the 100-ns transient spectrum both correspond to the first ground-state photo-intermediate. I was also surprised by the assignment of the 4.4 μ s kinetics to the conformational change of His149 (conclusion 2 above). Since at 10 ns, 60 % of His149 is off-like and 40 % on-like, I would rather think that the timescale of the movement is \sim 10 ns. I recommend performing additional transient absorption measurements in the 40 ps-100 ns time window.

Other major points:

- p1, title: I find the adjective "femtosecond" misleading, since the manuscript deals with ns and longer timescales. I recommend to remove it.

- p3, second paragraph: The description of previous literature is not fully appropriate. There is actually no clear signature of the cis protonated intermediates of Dronpa and Dronpa 2 in differential IR spectroscopy (check Warren et al., Nat. Commun. 2013, 4, 1461; Lukacs et al., J. Phys. Chem. B 2013, 117, 11954; Kaucikas et al., J. Phys. Chem. B 2015, 119, 2350). In particular, the blue-shift of the chromophore C=O stretching mode expected to occur upon

trans→cis isomerization is not observed. According to Kaucikas et al., this is due to the distorted structure of the trans chromophore in the off state. Femtosecond IR therefore only allows to exclude ESPT as the first photoswitching step. The evidence for the cis configuration of the first ground-state intermediate comes from (polarized) transient absorption spectroscopy (Yadav et al., J. Phys. Chem. B 2015, 119, 2404 and Colletier et al., J. Phys. Chem. Lett. 2016, 7, 882 - which should both be referenced in this paragraph). μ s chromophore deprotonation was also established by transient absorption spectroscopy - the same two references should be cited here.

- p5, results section and Figure 2b: "two positive maxima at 390 nm and 460 nm". The 460 nm band is not usual in a protonated fluorescent protein. Is there some anionic contribution remaining at pH 4 ? I suggest that you show in the supplement the steady-state spectra used to calculate this difference and comment on the 460-nm band. Did you perform an acid-base titration to determine pKa(s) ?

- p6, second paragraph: Why is chromophore deprotonation biexponential ? The spectral changes associated to the two components are rather different (see DADS in Figure 2c). Are there two successive proton transfer steps - in which case it may not be appropriate to calculate an average time constant for proton transfer -, or two chromophore or protein conformations that are sufficiently different to explain the spectral differences ? You should comment on this.

- p6, third paragraph and Supplementary Figure 2: The data are very noisy and I do not see the three components. Did you try with one or two ?

- p10, second paragraph and p12, second paragraph: I do not agree with the assignment of the 4.4 μ s phase to the conformational change of His149 (see above) - unless it is highly multiexponential, but this should be discussed in more detail.

- p12, second paragraph: "After excited-state decay, the first photoproduct in the ground-state is a cis protonated chromophore". Transient absorption data between 40 ps and 100 ns are lacking, which makes this conclusion questionable (see above).

- p14, first paragraph: "Woodhouse et al., to be described elsewhere". Does this mean that the experiments cannot be reproduced based on the indications given in the manuscript and the available literature ?

- p14, second paragraph: The level of detail is not sufficient to enable the reader reproducing the experiments. Please provide the original reference describing the transient absorption setup. I checked ref 42 but it cites another reference. Precise the size of the pump spot and that of the 488-nm LED spot on the sample so that photon fluxes can be calculated. Over how many pump shots did you average ? I understand that you used two lamps to convert the protein back to the off state between pump shots, could you provide more details about the first one ? What conversion percentage did you obtain this way ? How did you reconstruct spectra ? Did you correct from laser fluctuations ? What are the residuals of your fits ?

Minor points:

- p3, ref 8 should actually be ref 9 (photoswitching quantum yields of Dronpa)
- p4, I would replace ref 18 by Henderson et al., PNAS 2007, 104, 6672 which contains the on and off state structures of mTFP0.7, while ref 18 contains the on state structure of mTFP1
- p4, third paragraph: "on a time scale from a few ps to (μ s) ms" (there is a 2 ms phase in IrisFP, see Colletier et al., J. Phys. Chem. Lett. 2016, 7, 882)
- p5, results section and in several other places throughout the manuscript: replace "magenta" by "purple"
- p6, second paragraph: "the 4.4 μ s time constant is mainly characterized by a growth of the positive band at 460 nm - and a decay at 390 nm".
- p12, second paragraph: "Chromophore deprotonation occurs with (a characteristic) an average time constant of 759.1 μ s".
- p18, second paragraph and in the caption of Supplementary Figure 5: I think you should

exchange the 40 and 60 %.

- Figure 5b: The distance between the NE2 of His149 off-like and the phenol oxygen of the chromophore is lacking.

Reviewer #2 (Remarks to the Author):

The major goal of this paper is to fill in the later (ns) "on" events of the photoswitching behavior of a GFP variant that the group has previously characterized at the ps timescale (in a great paper in Nature Chemistry earlier this year). The major discovery is that the chromophore binding pocket has surprising heterogeneity on this timescale that differ from the traditional "on" structure. This conclusion is reached after analysis of maps created by various manipulations of the structure factors. The observed heterogeneity is likely linked to the protonation of the chromophore, which cannot be revealed directly here, but is nicely inferred through an analysis of distances between heavy atoms. The authors also discover new heterogeneity of the "off state" through the use of phenix.ensemble_refinement. I have three remaining issues with the paper that would benefit from further clarification:

As with reviewing their last paper in Nature Chemistry, I have some minor issues with the use of extrapolated structure factors and difference refinement. The use of extrapolated structure factors as a discovery/model building tool is appropriate, but given the high occupancy (40-50%), I don't understand why the difference refinement is preferred over refining a traditional multiconformer model (for example, assigning A, B altconfs to the intermediate and C, D, E altconfs to the ground state)

The presence of Trans2 in the off state structure is fascinating. If they subject either previous SFX (from the Nature Chemistry paper) or traditional synchrotron OFF datasets do they detect similar heterogeneity from ensemble refinement or a closer inspection of the map? Similarly, is there any evidence for His149 heterogeneity identified in Figure 5B in previous longer timescale "on state" datasets?

I think the authors are missing an opportunity if they leave their discussion of the structural models and the energy diagram of Fig 6 unlinked. Help me understand it by integrating them together in a proposed schematic that unifies what we learned on the ps and ns timescales! In particular, Is the new trans2 related to the model T intermediate of the last paper?

I review non-anonymously, James Fraser (UCSF). I further suggest that the authors publicly post their work by depositing the manuscript on a preprint server (e.g. BioRxiv) immediately. This will allow the rest of the world have an opportunity to help improve their manuscript prior to a "final version" appearing upon the completion of formal peer review. Additionally, this practice is allowed by Nature Communications.

Reviewer #3 (Remarks to the Author):

This manuscript by Woodhouse et al. aims at addressing the mechanism of reversible photoswitching in GFP-type proteins. It is well established that the process involves both chromophore isomerization and a change of the chromophore protonation state. In recent years, evidence has mounted that reversible photoswitching is a multistep process in either direction, with isomerization in the excited state followed by proton transfer in the ground state.

By combining time-resolved serial crystallography at an X-ray free electron laser and pump-probe time-resolved UV-visible spectroscopy, Weik and co-workers have studied the on-switching of rsEGFP2, i.e., the transition from a non-fluorescent, neutral trans chromophore (off-state) to a fluorescent, anionic cis chromophore (on-state), in great detail. First results on excited-state events on the ps timescale have been published by the authors earlier in Nature Chemistry (ref 20 of the present paper). Here, they have followed up on this work and focused on processes on the nanosecond to microsecond time scales.

The spectroscopic data (collected at 100 ns – 10 ms) suggest that the first ground state intermediate is a neutral (protonated) cis chromophore, which then deprotonates on the μs time scale to yield the final on-state. The X-ray structure determined 10 ns after the photoswitching laser pulse (laser_on_Δ10ns) shows multiple chromophore conformations and also multiple conformations of other residues, notably His149. After comparing these conformations with those present in the 3D structures of on-state and off-state rsEGFP2 from conventional x-ray crystallography, the authors felt reassured that the first ground state intermediate is indeed a neutral cis chromophore.

The spectroscopic data are straightforward and clearly support the key claim of the paper; i.e., excited-state isomerization followed by ground state deprotonation. The SFX data and their refinement are absolutely state-of-the-art, and a great deal of effort has gone into the refinement of multiple conformations (trans1, trans2, cis). Most pronounced are the changes of His149. Unfortunately, the nature of the proton pathway, which would be important for the mechanism, as well as the observed trans-state heterogeneity have remained elusive.

The manuscript is a bit difficult to grasp, partially because of the nomenclature. Also, clear depictions of the irradiation protocols would be helpful. Furthermore, the manuscript could benefit from better working out the new insights gained from the data for the mechanism. The model in Figure 6 does not go beyond a description of the kinetics as measured by spectroscopy. Therefore, the last sentence of the abstract (..establishing a detailed mechanism..) promises more than the present version of the manuscript delivers.

Specific remarks:

Page 6: There are two processes showing a significant KIE, with time constants of 40 and 919 μs , which were population averaged to 759.1 μs . The processes differ by more than an order of magnitude in time. Why should one subsume kinetically so different processes into one process - just for simplicity?

Page 12: I find the second half of the second paragraph (starting with: Transient absorption...) rather unclear. I would recommend elaborating on this issue by using a schematic.

Page 14, top paragraph: "Woodhouse et al., to be described elsewhere." All required procedures should be included in this work.

Page 15, bottom paragraph: I cannot find a Fig. 8 in Ref 22.

We are grateful to all three referees for their thoughtful, detailed and constructive comments and criticisms. We have carried out the additional spectroscopic experiments (femtosecond transient absorption spectroscopy in the 40 ps to 2 ns window) suggested by referee 1. We have repeated all nanosecond transient absorption spectroscopy experiments and averaged a sequence of eight single-pulse experiments (only one single-pulse experiment in the submitted version), leading to a higher signal-to-noise ratio so that the gap between 10 and 100 ns could be filled. The additional experiments are described in new supplementary figures S1, S2 and S9. On the crystallographic side, we have double checked the original data processing and discovered and corrected an error during the merging stage. The overall dark and light models did not change, but we now model His149 in the 10-ns intermediate structure in only one conformation (*off*-like), instead of two (*off*-like and *on*-like) in the submitted version. Following reprocessing occupancies of the three conformers in the laser_off model also changed slightly (65, 10, 25%, as compared to 70, 10, 20% in the submitted version). In order to further detail the switching mechanism, we have extended figure 6 and added a discussion on three potential proton pathways.

In the following, we reproduce the referee comments in blue. Our responses are in black, including additional figures and references when necessary, and text modified in the revision is in italics.

Reviewer #1 (Remarks to the Author):

In my opinion, the manuscript by M. Weik and collaborators will be of interest to the community of spectroscopists and structural biophysicists studying light-induced reactions in fluorescent proteins. However I do not think that it will change thinking in this field, since most of the conclusions are not new (see below for more detail). In addition, data presented in the manuscript currently have two major weaknesses that question the author's conclusions (see below). I think this should be solved before the manuscript can be published.

First of all, we would like to thank the referee for his/her very positive comments on the importance of our results to scientists studying photo-reactions in fluorescent proteins. We agree with the referee that this is interesting. We believe that this is even more so true that a manuscript has been published (Laptenok et al (2018) (*Nature Chemistry* DOI 10.1038/s41557-018-0073-0, 06/11/2018) while our study was under review that questioned again the now widely accepted excited-state isomerisation and deprotonation ground state mechanism for Dronpa, the most studied representative Anthozoa RSFP. Concerning RSFPs from Hydrozoa (e.g. jellyfish), both structural and kinetic details of ground state processes are to be determined. Here, we studied *off*-to-*on* photoswitching in rsEGFP2 (Hydrozoa origin), a negative RSFP that is a key marker in super-resolution RESOLFT microscopy.

We also thank the referee for his/her effort and time to carefully examine our study and to provide an extensive list of constructive comments on our work. We address all issues raised in detail below, and are grateful for the opportunity to outline how we have solved the two major weaknesses identified.

Main results and conclusions:

The manuscript by M. Weik and collaborators is the follow-up of a manuscript published earlier this year by the same authors (Coquelle et al., Nat. Chem. 2018, 10, 31). Both manuscripts investigate the mechanism of off→on photoswitching in the reversibly photoswitchable fluorescent protein (RSFP) rsEGFP2 by a combination of XFEL-based time-resolved crystallography and transient absorption spectroscopy. While the first manuscript dealt with primary events in the excited-state (ps

timescale), the present one focuses on slower, ground-state reaction steps (from 10 ns to milliseconds).

The main new data are the following:

- The authors solved the structure of rsEGFP2 10 ns after excitation of the off state. The 10-ns structure features a *cis* chromophore, which indicates that *trans*→*cis* isomerization is completed in this time-scale. The protein layout around the chromophore is different from that observed in the final on state, and shows some heterogeneity with 2 conformations of His149 (40 % on-like, 60 % off-like).

- The structure of the initial off state was also reexamined. Surprisingly, it was found to consist of two different *trans* conformers : *trans*1, corresponding to the off state structure reported in Coquelle et al. (and in El Khatib et al., *Sci. Rep.* 2016, 6, 18459) and *trans*2, a hitherto unreported conformer accounting here for 20 % of the population.

- Transient absorption spectra were measured from 100 ns to milliseconds, following excitation of the off state. They reveal 3 kinetic phases : 4.4 μ s, 40 μ s and 919 μ s, with kinetic isotope effects of 1.8 and 2.4, resp., for the two longest components upon hydrogen-deuterium exchange.

The authors conclude from these data 1) that the first ground-state intermediate in rsEGFP2 off→on photoswitching is a *cis* protonated form corresponding to the structure solved at 10 ns, 2) that the 4.4 μ s component corresponds to the movement of His149 from its off-like to its on-like conformation, and 3) that chromophore deprotonation occurs with a characteristic time of 759 μ s (average of the 40 μ s and 919 μ s components), and is accompanied by a further migration of His149 towards its final on state position.

Novelty and interest:

As far as techniques are concerned, XFEL-based time-resolved crystallography is an exciting new technique that gives access to the light-induced structural dynamics of proteins on time-scales as short as ~100 fs. In Coquelle et al., the authors reported structures of rsEGFP2 at 1 ps and 3 ps. In the present manuscript however, they report a structure at 10 ns, a timescale that has already been accessible for ~20 years with synchrotron-based time-resolved crystallography (see for instance the 1-ns structure of the photoactive yellow protein in Perman et al., *Science* 1998, 279, 1946). But this is the first nanosecond structure of a RSFP, and therefore I expect that it will be of interest to RSFP specialists.

Concerning the author's conclusions (see list above):

- The *cis* protonated nature of the first ground-state intermediate in RSFP off→on photoswitching (point 1) was previously established by Yadav et al. (*J. Phys. Chem. B* 2015, 119, 2404) in the case of Dronpa and Dronpa-2, and by Colletier et al. (*J. Phys. Chem. Lett.* 2016, 7, 882) for a third RSFP, IrisFP. Parallel investigations by femtosecond IR spectroscopy fueled some doubts because they failed to detect the IR signature expected for a *cis* chromophore (Warren et al., *Nat. Commun.* 2013, 4, 1461; Lukacs et al., *J. Phys. Chem. B* 2013, 117, 11954). But it is now established that the lack of a clear signature for the *cis* chromophore in differential IR spectroscopy is due to the distorted structure of the *trans* chromophore in the off state (Kaucikas et al., *J. Phys. Chem. B* 2015, 119, 2350).

We agree with the referee that spectroscopic evidence for the first ground-state intermediate in RSFP off→on photoswitching being of *cis* protonated nature was previously reported for an RSFP of Anthozoan origin (Dronpa, Dronpa-2 and IrisFP). However, a study published while our manuscript was under review questions again this *cis* protonated nature (Laptenok

et al (2018) (*Nature Chemistry* DOI 10.1038/s41557-018-0073-0, 06/11/2018) so that the jury is still out concerning RSFP of Anthozoan origin. As to RSFPs from Hydrozoa (e.g. jellyfish), both structural and kinetic details of ground state processes are to be determined and were, to our knowledge, not reported. Referring to our earlier work (Coquelle et al., 2018) Laptенок and co-workers write that "The structure of the distorted state dA2 cannot be further resolved by infrared spectroscopy, but is an important target for both femtosecond crystallography and quantum-mechanical calculations. The very recent femtosecond crystallography data recovered a distorted chromophore structure formed within 3 ps, although that was assigned to an excited state species; extensions to later times will be significant." The present work provides such an extension to later times and on Hydrozoa RSFP.

The results obtained on Hydrozoa RSFP do not match with the ground-state isomerization process proposed in Laptенок et al. for Anthozoa RSFP. In the current manuscript, we provide a structural proof that the first ground-state intermediate in Hydrozoa RSFP off→on photoswitching is indeed of *cis* protonated nature for rsEGFP2.

We have modified the relevant sentences in the introduction of the revised version as follows :

“Later, however, picosecond time-resolved infra-red (TR-IR) spectroscopy on Dronpa (Warren, et al 2013) and its fast-switching M159T mutant (Kaucikas et al 2015) indicated that isomerization occurs during picosecond excited-state decay. This has also been suggested by femtosecond UV-visible transient anisotropy absorption spectroscopy, which attributed the first photoproduct to a cis-protonated chromophore (Yadav et al 2015). The same study also showed that chromophore deprotonation occurs in the ground state on the microsecond time scale. However, another TR-IR study on Dronpa-M159T mutant advocated that both isomerization and deprotonation are ground-state processes, attributing the ps spectroscopic changes in the excited state to protein conformational changes priming the chromophore for switching (Lukacs et al 2013). This view was corroborated by a follow-up TR-IR study on Dronpa-M159T involving isotope labelling (Laptенок et al 2018). In that study, evidence has been provided for chromophore distortion and excited-state decay on the picosecond time scale, followed by formation of the cis protonated chromophore with a 91 ns time constant before the final deprotonation step. Importantly, the structure of the cis-protonated switching intermediate in Dronpa has remained elusive.”

- Point 2 (conformational movement of His149) is new and illustrates the interplay between chromophore and protein conformational changes in course of RSFP photoswitching. However, see my comments on timescales below.

- μs-ms chromophore deprotonation (point 3) was previously reported by Yadav et al. in Dronpa and Dronpa-2 and by Colletier et al. in IrisFP.

With the exception of point 2 which is new and will be of interest to RSFP specialists, the work by M. Weik and collaborators essentially shows that rsEGFP2 has a similar off→on photoswitching mechanism as Dronpa, Dronpa-2 and IrisFP. Since rsEGFP2 is a variant of GFP from jellyfish while Dronpa, Dronpa-2 and IrisFP are all variants of fluorescent proteins from corals, this conclusion will also be of interest to specialists.

As outlined above, the latest publication on Dronpa (Anthozoa RSFP, Laptенок et al., 2018) suggests a more complex mechanism than excited-state isomerization followed by ground state deprotonation so that the photoswitching mechanism in Anthozoa RSFP is still controversially debated. As to rsEGFP2, we are confident that our manuscript firmly establishes the photoswitching mechanism in this Hydrozoa RSFP. Further work on Dronpa will clarify similarities and differences in the switching mechanisms of Hydrozoa and Anthozoa RSFPs.

Major points that question the conclusions:

In my opinion, the two following problems question the manuscript's conclusions and should therefore be solved in priority:

- When reading the manuscript, I was very embarrassed by the inconsistency between the new off state structure and that reported a few months ago by the same authors in Coquelle et al. (and in 2016 in El Khatib et al.). The new, previously unreported *trans2* conformer corresponds to 20 % of the population, this is more than significant. Yet the three structures have similar resolutions (1.7 Å for the present one and that in Coquelle et al., 1.5 Å for El Khatib et al.). Where does this lack of reproducibility come from ? The authors do not give any explanation. In my opinion, the fact that the initial state of the reaction under study is ill-defined raises doubts over the entire work. This point definitely has to be clarified before the manuscript can be published.

We agree with the referee that the occurrence of the new *trans2* conformation in the *off*-state of the chromophore at 20% (now 25%, see above) occupancy, in addition to the well-described *trans1* conformation at 70% (now 65%) occupancy, is puzzling.

Indeed, no such additional conformation has been reported in our earlier *off*-state structures determined by traditional cryo-crystallography at a synchrotron (El Khatib et al 2016) and by SFX at room temperature at an XFEL (Coquelle et al 2018). Referee 2 suggested analyzing those two structures by the same ensemble-refinement approach as the one presented in Figure 3 so that a low-occupancy *trans2* state can be identified, if present. We had already carried out such an analysis (see result in Figure R1a and b below). Analysis of the *off*-state data collected by cryo-crystallography (Fig. R1a) did not provide evidence of the existence of a *trans2* conformation for the chromophore, nor did it provide evidence that an alternative His149 conformation is present. Likewise, analysis of the *off*-state data collected by SFX (Fig. R1b) did not reveal the existence of *trans2* or any alternative His149 conformation (Fig. 3). Consequently, *trans2* had not been included in the *off*-state structures determined by SFX (Coquelle et al 2018) and by cryo-crystallography at a synchrotron (El Khatib et al 2016). Figure R1 has been added as an additional supplementary figure (Fig. S8) in the revised version of the manuscript.

Figure R1: Result of ensemble refinements against (a) the *off*-state data in El Khatib et al 2016) starting for the *off*-state model (chromophore 100% *trans1*, PDB code 5DTY) and (b) the *laser_off* data set reported in Coquelle et al 2018 starting for the *off*-state model (PDB code 5O8A) from which the *cis* alternate conformation was omitted so that the chromophore was fully occupied in the *trans1* conformation.

[redacted]

Interestingly, a recent manuscript (Chang et al (2019) JACS) also provides evidence for the existence of a *trans2* chromophore in cryo-crystallographic synchrotron experiments. Illumination at 488 nm of an rsEGFP2 variant with a substituted phenolate ring yielded *trans2* when crystals with a contracted unit cell have been examined, whereas the *trans1* conformation is populated when crystals with a larger unit cell have been used. The authors suggest the tighter packing of proteins in the contracted crystal reduces the flexibility of the chromophore pocket, thereby limiting the volume available for the isomerizing chromophore and favoring the volume-conserving *cis* > *trans2* transition, rather than the volume-consuming *cis* > *trans1* transition. We refer to this manuscript in the Discussion section:

‘Evidence for the existence of a trans2 chromophore also comes from cryo-crystallographic synchrotron experiments, in which illumination at 488 nm of an rsEGFP2 variant with a substituted phenolate ring yielded trans2 when crystals with a contracted unit cell were examined, whereas the trans1 conformation is populated when crystals with a larger unit cell were used (Chang et al 2019). The authors suggest the tighter packing of proteins in the contracted crystal reduces the flexibility of the chromophore pocket, thereby limiting the volume available for the isomerizing chromophore and favoring the volume-conserving cis > trans2 transition, rather than the volume-consuming cis > trans1 transition. Coincidentally, different illumination conditions were used for the two datasets.’

Our approach to refinement of the 10-ns intermediate-state structure in the current manuscript is based on extrapolated structure factors that only analyse differences between the illuminated and dark structures. Therefore, any *off*-state heterogeneity in the *off*-state structure is cancelled out. In addition, since the 10-ns intermediate structure is occupied at the 50% level in the laser-on structure, we can exclude that it forms from *trans2* that is only occupied at 25% in the laser-off structure before photoswitching with 400 nm light. Yet, it is possible that a fraction of the 10-ns intermediate structure originates from switched *trans 2* conformers. We argue that none of the three main findings in the present work (*cis*-protonated nature of the first ground-state intermediate; interplay between chromophore and protein conformational changes during photoswitching; μ s-ms chromophore deprotonation) are questioned by the presence of *trans2* in addition to *trans1* so that the main conclusions of the manuscript remain valid. [redacted]

The second major weakness is that there is no temporal match between crystallographic and spectroscopic data, which results in some vagueness in the

timescales of rsEGFP2 photoswitching. The structure of the intermediate was taken at 10 ns, while transient absorption was measured from 100 ns to milliseconds. There is also a gap of more than three orders of magnitude with the transient absorption data published in Coquelle et al., which end at 40 ps. The 100-ns spectrum shown in the present manuscript (Figure 2a) is very different from the 40-ps spectrum shown in Coquelle et al. (Supplementary Figure 1c), which suggests that kinetic phases exist for rsEGFP2 in the 40 ps-100 ns time window. These kinetic phases have not been characterized. Yet, the authors claim that the 10-ns structure and the 100-ns transient spectrum both correspond to the first ground-state photo-intermediate. I recommend performing additional transient absorption measurements in the 40 ps-100 ns time window.

We followed the referee's request to collect time-resolved absorption spectroscopy data in the 40 ps to 100 ns time window. In particular, we carried out femtosecond transient absorption spectroscopy in the 40 ps to 2 ns window (new Supplementary fig 1). We found the existence of an additional kinetic phase in the ground state with a characteristic time of about 87 ps. Furthermore, we repeated all nanosecond transient absorption spectroscopy experiments presented in the submitted version and averaged a sequence of eight single-pulse experiments (only one single-pulse experiment in the submitted version). The 8-fold repetition lead to a higher signal-to-noise ratio so that the gap between 10 and 100 ns could be filled (new Supplementary fig 2). No spectral change was observed in the 10 to 100 ns time window, so that the 10 ns intermediate-state structure indeed corresponds to the one of a protonated *cis* chromophore. Figures 2 and 6 and Supplementary figures 3 and 5 have been updated and the relevant text in the Results section has been modified as follows:

*“Time-resolved UV-visible transient absorption spectra were recorded from 40-ps to 10-ms after excitation of rsEGFP2 in its off-state in solution (50 mM HEPES pH 8, 50 mM NaCl; Fig. 2, Supplementary Fig. 1). On the examined time-scale, the excited state had decayed (Coquelle et al 2018) so that the photodynamics of rsEGFP2 in the ground state was monitored. From 40 ps to 2 ns (Supplementary Fig. 1a), transient spectra show a negative and a positive band at 420 and 460 nm, respectively, evolving with a characteristic time constant of $87 \text{ ps} \pm 8 \text{ ps}$ (Supplementary Fig. 1c). A negative band at 500 nm originates from stimulated emission from residual 10% of rsEGFP2 that had remained in the *cis* anionic on state (Supplementary Fig. 1b) and vanished after 2 ns (Supplementary Fig. 1d). Between 10 and 100 ns, there are no significant spectral evolutions (Supplementary Fig. 2).”*

New Supplementary Figure 1: Time-resolved difference absorption spectra recorded between 40 ps and 2 ns after a 400 nm femtosecond excitation of the *trans* protonated *off* state (a) and the *cis* anionic *on* state (b) of rsEGFP2 in H₂O solution (50 mM HEPES pH 8, 50 mM NaCl). The spectrum without laser excitation was subtracted to calculate the difference spectrum. The grey arrow (a) indicates the disappearance of the 420 nm band within 87 ps. Kinetic traces at 420 nm (c) and 500 nm (d) extracted from panels a and b are shown.

New Supplementary Figure 2: Time-resolved difference absorption spectra recorded between 10 and 100 ns after a 410 nm nanosecond excitation of the *trans* protonated *off* state of rsEGFP2 in H₂O solution (50 mM HEPES pH 8, 50 mM NaCl). The spectrum without laser excitation was subtracted to calculate the difference spectrum.

I was also surprised by the assignment of the 4.4 μ s kinetics to the conformational change of His149 (conclusion 2 above). Since at 10 ns, 60 % of His149 is off-like and 40 % on-like, I would rather think that the timescale of the movement is \sim 10 ns.

After reprocessing the crystallographic data (see above), the electron density maps were of better quality and His149 was modeled in only one (*off*-like) and not two (*off*-like and *on*-like) conformations. As a consequence, we now suggest that the His149 transition from the *off*-like conformation at 10 ns to its final *on* conformation in the *cis* anionic state either corresponds to the 5.6 μ s process (see extended figure 6) or occurs during chromophore deprotonation at a later stage.

We have modified the relevant Discussion paragraph as follows:

'Transient absorption spectroscopy in H₂O solution evidences a further ground-state process with a time constant of about 5.57 μ s (no isotope effect) that might correspond to a protein rearrangement during which the off-like conformation of His149 transitions to the on conformation (Fig. 6). Alternatively, the conformational change in His149 could accompany chromophore deprotonation that occurs with characteristic time constants of 36.1 and 825 μ s (isotope effect) to yield the final cis anionic on-state.'

Other major points:

- p1, title: I find the adjective "femtosecond" misleading, since the manuscript deals with ns and longer timescales. I recommend to remove it.

The term 'femtosecond' in 'serial femtosecond crystallography' refers to the XFEL pulse length and not to the pump-probe delay. The technique has been named in the paper describing its second successful application by Boutet et al 2012, entitled: High-resolution protein structure determination by serial femtosecond crystallography. We thus need to keep the title as it is.

- p3, second paragraph: The description of previous literature is not fully appropriate. There is actually no clear signature of the cis protonated intermediates of Dronpa and Dronpa 2 in differential IR spectroscopy (check Warren et al., Nat. Commun. 2013, 4, 1461; Lukacs et al., J. Phys. Chem. B 2013, 117, 11954; Kaucikas et al., J. Phys. Chem. B 2015, 119, 2350). In particular, the blue-shift of the chromophore C=O stretching mode expected to occur upon trans \rightarrow cis isomerization is not observed. According to Kaucikas et al., this is due to the distorted structure of the trans chromophore in the off state. Femtosecond IR therefore only allows to exclude ESPT as the first photoswitching step. The evidence for the cis configuration of the first ground-state intermediate comes from (polarized) transient absorption spectroscopy (Yadav et al., J. Phys. Chem. B 2015, 119, 2404 and Colletier et al., J. Phys. Chem. Lett. 2016, 7, 882 - which should both be referenced in this paragraph). μ s chromophore deprotonation was also established by transient absorption spectroscopy - the same two references should be cited here.

We agree with the reviewer that our attempt to simplify the situation led to misunderstandings. We modified the relevant section by adding the results of Laptinok et al. 2018 and by specifying that Yadav et al 2015 showed that chromophore deprotonation occurs in the ground state on the microsecond time scale:

"The first investigation of Dronpa by ultrafast optical spectroscopy suggested that the deprotonation of the trans-phenol occurs in the excited state on the ps timescale via an excited-state proton transfer (ESPT) mechanism (Fron et al 2007). Later, however, picosecond time-resolved infra-red (TR-IR) spectroscopy on Dronpa (Warren et al 2013) and its fast-switching M159T mutant (Kaucikas et al 2015) indicated that isomerization occurs

during picosecond excited-state decay. This has also been suggested by femtosecond UV-visible transient anisotropy absorption spectroscopy, which attributed the first photoproduct to a cis-protonated chromophore (Yadav et al 2015). The same study also showed that chromophore deprotonation occurs in the ground state on the microsecond time scale. However, another TR-IR study on Dronpa-M159T mutant advocated that both isomerization and deprotonation are ground-state processes, attributing the ps spectroscopic changes in the excited state to protein conformational changes priming the chromophore for switching (Lukacs et al 2013). This view was corroborated by a follow-up TR-IR study on Dronpa-M159T involving isotope labelling (Laptenok et al 2018). In that study, evidence has been provided for chromophore distortion and excited-state decay on the picosecond time scale, followed by formation of the cis protonated chromophore with a 91 ns time constant before the final deprotonation step. Importantly, the structure of the cis-protonated switching intermediate in Dronpa has remained elusive.”

- p5, results section and Figure 2b: "two positive maxima at 390 nm and 460 nm". The 460 nm band is not usual in a protonated fluorescent protein. Is there some anionic contribution remaining at pH 4 ? I suggest that you show in the supplement the steady-state spectra used to calculate this difference and comment on the 460-nm band. Did you perform an acid-base titration to determine pKa(s) ?

Determination of steady state spectra and pKa with acid-base titration were reported earlier (El Khatib et al 2016). The origin of the maxima at 460 is not clear because at pH 4 the protein starts to denature. We thus decided to remove steady state spectra at pH 4 in the Figure 2 and any discussion about it in the manuscript.

“Figure 2 shows spectral changes from 100 ns to 9 ms. The transient difference absorbance spectrum at 100 ns shows a broad positive band with a maximum at 400 nm (dark blue in Fig. 2a). This band evolves within 4.9 μ s to a spectrum (light blue) with two positive maxima at 390 nm and 460 nm. Then within 90.6 μ s, the first peak (390 nm) vanishes, while a negative band appears at 410 nm. The second peak (460 nm) increases and shifts to 480 nm (Fig. 2b). Subsequently, and on a timescale from 100 μ s to 9 ms, the maximum of the negative band shifts from 420 to 400 nm while the positive band at 480 nm further increases in amplitude. The 480 nm band is characteristic of the cis anionic form (Grotjohann et al 2012).”

New Figure 2: Transient absorption spectroscopy of rsEGFP2 in H₂O solution (50 mM HEPES pH 8, 50 mM NaCl). Time-resolved difference absorption spectra recorded after a 410 nm nanosecond excitation of the *trans* protonated *off* state in the time windows from 100 ns to 9 ms (*a* - *c*). The spectrum without laser excitation was subtracted to calculate the difference spectra. The coloured arrows (red in (*a*), cyan in (*b*) and light green in (*c*)) correspond to the three time constants (5.57, 36.1 and 825 μ s, respectively) identified by a global fit analysis of kinetic traces for all wavelengths. (*d*) Decay associated spectra obtained by fitting the kinetic traces in panels *a* – *c* for all wavelengths with the sum of three exponential functions.

- *p6, second paragraph: Why is chromophore deprotonation biexponential ? The spectral changes associated to the two components are rather different (see DADS in Figure 2c). Are there two successive proton transfer steps - in which case it may not be appropriate to calculate an average time constant for proton transfer -, or two chromophore or protein conformations that are sufficiently different to explain the spectral differences ? You should comment on this.*

The nature of these two time constants is unclear. The DADS for the 36 μ s component is characteristic of a band shift, while the 825 μ s time evolution is characterized by a clear isosbestic point. The magnitude of the isotope effect is almost equal for 36 μ s and 825 μ s processes ($k_H/k_D = 2.45$ and 2.47 , respectively). This supports that the time-constants characterise the same deprotonation process in one protein conformation. The results section was modified as follows:

“Global fit analysis of kinetic traces for all wavelengths identified three time constants of $5.57 \pm 0.02 \mu$ s, $36.1 \pm 0.1 \mu$ s and $824.8 \pm 0.3 \mu$ s, respectively. When similar experiments were carried out in D₂O solution (50 mM HEPES pD 8, 50 mM NaCl ; Supplementary Fig. 3), time evolution also required fitting with three exponential functions, yielding time constants of $5.16 \pm 0.02 \mu$ s, $88.4 \pm 0.2 \mu$ s and $2041.1 \pm 0.7 \mu$ s. Thus, the first time constant is similar in H₂O and D₂O, but a significant isotope effect is observed for the two others ($k_H/k_D = 2.45$ and 2.47 , respectively) which can be assigned to proton transfer steps. Decay associated spectra

(Fig. 2d) show that the 5.57 μs time constant in H_2O solution is mainly characterized by a growth of the positive band at 460 nm and a decay at 390 nm. The 36.1 μs time constant has some positive and negative contributions characteristic of the band shift observed, while the 825 μs time constant is mainly characterized by the respective decay and growth of the 390-nm and 480-nm bands.’

”

New Figure 6: Model for the rsEGFP2 *off*-to-*on* photoswitching process. The bold purple bar represents the *laser_on_Δ10ns* intermediate structure determined by TR-SFX. Time constants correspond to those determined by femtosecond (87 ps) and nanosecond-resolved pump-probe UV-visible absorption spectroscopy (5.57, 36.1 and 825 μs) in H_2O solution (50 mM HEPES pH 8, 50 mM NaCl). All three *cis* protonated states (purple bars) are ground-state intermediates, interconversion of which involves rearrangements of the protein matrix (orange arrows). Corresponding structural models for the chromophore and His149 as determined by X-ray crystallography are shown in panels *a*, *b* (this work) and *d* (Coquelle et al 2018). *Trans1* and His149-*off* (light grey) and *trans2* and His149-*supp* (dark grey) of the *trans* protonated *off*-state (*a*, Figure 3d) transitioned to *cis* and His149-*off-like* at 10 ns (*b*, Figure 4b, Figures 5b,e). His149-*off-like* has moved to His149-*on* in the *cis* anionic *on* state (*d*, Figures 5c,f). The structure whose intermediate forms in 5.6 μs remains elusive (*c*).

- p6, third paragraph and Supplementary Figure 2: The data are very noisy and I do not see the three components. Did you try with one or two ?

We agree with the reviewer that spectroscopic data for the crystals and the decay associated spectra in the region between 350 and 440 (Supplementary Figure 2 in submitted version, Supplementary Figure 4 in the revised version) are very noisy. However, when the time traces

at 480 nm are plotted (Figure R7, not shown in the manuscript), it is clear that three time constants are needed to correctly fit the data. In addition the three time constants retrieved for the microcrystal suspension ($4.75 \pm 0.13 \mu\text{s}$, $42.9 \pm 0.8 \mu\text{s}$ and $295 \pm 2 \mu\text{s}$) are similar to the one obtained for rsEGFP2 in ammonium sulfate solution ($4.23 \pm 0.02 \mu\text{s}$, $40.4 \pm 0.2 \mu\text{s}$ and $245.2 \pm 0.2 \mu\text{s}$), which reinforces the choice of a three-exponential model for microcrystals data. The reviewer may note that the time constants extracted from the microcrystal measurements are slightly different in the revised and the submitted version (4.75 vs $2.6 \mu\text{s}$, 42.9 vs $36.5 \mu\text{s}$ and 295 vs $310 \mu\text{s}$). The difference is due to the use of different software packages for global decay fitting (*Origin* and *Igor* for the submitted, only *Igor* for the revised version).

Figure R7: Kinetic time traces between 0 and 100 μs (a) and between 0 and 2000 μs (b) at 480 nm recorded after 410 nm nanosecond pulse excitation of a suspension of rsEGFP2 microcrystals in 100 mM HEPES pH 8, 2.5 M ammonium sulphate.

p10, second paragraph and p12, second paragraph: I do not agree with the assignment of the 4.4 μs phase to the conformational change of His149 (see above) - unless it is highly multiexponential, but this should be discussed in more detail.

See our explanations above.

- p12, second paragraph: "After excited-state decay, the first photoproduct in the ground-state is a cis protonated chromophore". Transient absorption data between 40 ps and 100 ns are lacking, which makes this conclusion questionable (see above). Missing data in the 40 ps – 2 ns (Supplementary Figure 1) and the 10 ns – 100 ns (Supplementary Figure 2) windows have now been obtained by additional femtosecond and nanosecond time-resolved absorption spectroscopy experiments, respectively (see above). We can now firmly conclude that the first photoproduct contains a cis protonated chromophore.

- p14, first paragraph: "Woodhouse et al., to be described elsewhere". Does this mean that the experiments cannot be reproduced based on the indications given in the manuscript and the available literature ?

Details given in the materials-and-methods section are sufficient to reproduce microcrystallization. Therefore, we omitted "Woodhouse *et al.*, to be described elsewhere" in the revised version.

p14, second paragraph: The level of detail is not sufficient to enable the reader reproducing the experiments. Please provide the original reference describing the transient absorption setup. I checked ref 42 but it cites another reference. Precise the size of the pump spot and that of the 488-nm LED spot on the sample so that photon fluxes can be calculated. Over how many pump shots did you average ? I understand that you used two lamps to convert the protein back to the off state between pump shots, could you provide more details about the first one ? What conversion percentage did you obtain this way ? How did you reconstruct spectra ? Did you correct from laser fluctuations ? What are the residuals of your fits ?

In the submitted version, only single-pulse experiments without averages were presented. To satisfy the reviewer request to obtain data in the 40 ps to 100 ns time window, we repeated all nanosecond transient absorption experiments with a modified setup and averaged eight single-pulse experiments to increase signal-to-noise ratio in the 10 ns to 100 ns time window (Figure 2, Supplementary Figure 3, Supplementary Figure 5). A reference describing the modified set up (Stoll, T. *et al.* [Rh(III)(dmbpy)₂Cl₂]⁺ as a highly efficient catalyst for visible-light-driven hydrogen production in pure water: comparison with other rhodium catalysts. *Chemistry* 19, 782-92 (2013), new reference 43) and as well as a new figure (Supplementary figure 9) have been added in the revised version. Quality of the fit was checked by analyzing the residuals (Figure R8, not shown in the manuscript). In the revised version, the materials-and-methods section now contains a much more detailed experimental description:

*“Femtosecond transient absorption spectroscopy from 40 ps to 2 ns was carried out on rsEGFP2 (12.5 mg/ml) in a H₂O solution containing 50 mM HEPES pH 8, 50 mM NaCl according to the protocol reported earlier (Coquelle *et al* 2018) (Supplementary Fig. 1).*

Nanosecond transient absorption spectroscopy experiments were carried out from 10 ns to 10 ms (Figure 2, Supplementary Figs. 2, 3, 5) using a conventional flash-photolysis setup (90° geometry between pump and probe beams, Supplementary Fig. 9) {Stoll, 2013 #78} on three rsEGFP2 solutions (46 μM) : i) 50 mM HEPES pH 8, 50 mM NaCl in H₂O, ii) 50 mM HEPES pH 8, 50 mM NaCl in D₂O, iii) 50 mM HEPES pH 8, 50 mM NaCl, 1.25 M (NH₄)₂SO₄ in H₂O. Initially, the solutions (2 ml) were illuminated for 3 min at 488 nm (Cobolt 06-MLD 488; 200 mW, beam diameter 1 mm) to photoswitch 90% of the proteins to the trans protonated off state (absorbance of about 1 at 410 nm for a 1 cm thickness). Pump beam (410 nm, 8 ns, 1.6 mJ, beam size approx. 1 mm × 10 mm) was focused using a cylindrical lens onto a spectroscopic cell (Hellma, chamber thicknesses 1 mm) containing the protein solution (Supplementary Fig. 9a). The time traces of the variation in absorbance were recorded from 350 to 520 nm in 10-nm steps (3.5 nm spectrometer bandpass) to reconstruct the transient difference absorption spectra at different time delays after the pump-pulse excitation. The recorded time traces were obtained by averaging a sequence of eight pump-probe experiments (laser energy fluctuation is below 0.1 mJ pulse-to-pulse and is not corrected). There was a time lapse of 30 s between two consecutive pump-pulse excitations. To photoswitch the proteins in the stationary solution back to the off-state after each 410 nm pump pulse the spectroscopic cell containing the stationary protein solution was illuminated continuously by 490-nm light (collimated Thorlabs LED M490LA, 200 mW, beam diameter 9 mm; Supplementary Fig. 9a). Time traces of the variation in absorbance can be analysed only at and above 10 ns, given the 8-ns pump pulse length. A transient difference absorption

spectrum at one time delay (t) is obtained by averaging 10 time delay points (from $t - 5$ points to $t + 5$ points) of the time traces. Three time delay windows were explored from 10 ns to 5 μ s (500 ps / point), from 5 μ s to 100 μ s (10 ns / point) and from 100 μ s to 10 ms (1 μ s / point). A global decay analysis was carried out on the raw data of time traces and analyzed using Igor Pro. The quality of the fit was checked by analyzing the residuals.

For spectroscopy experiments on microcrystal suspensions (Supplementary Fig. 4), rsEGFP2 microcrystals ($3 \times 3 \times 3 \mu\text{m}^3$) were suspended in 100 mM HEPES pH 8, 2.5 M ammonium sulphate. Suspensions were initially irradiated for 30 min at 488 nm (Cobolt 06-MLD 488; 200 mW, beam diameter 1 mm) to achieve off-switching. For the subsequent ns – ms transient absorption experiment, the microcrystal suspension was placed in a spectroscopic cell (Hellma, chamber thickness 1 mm, Supplementary Fig. 9b). The conventional flash photolysis setup was modified (15° between pump and probe beams, Supplementary Fig. 9b) to minimize light scattering. The pump laser (410 nm, 8 ns, 5 mJ, beam diameter approx. 2 mm) was focused using a spherical lens onto the cell containing the microcrystal suspension. The probe light-source was not modified from conventional experiments. The diameter of both pump and probe beams was 2 mm through the spectroscopic cell. Following single-shot excitation, the time traces of the variation in absorbance were recorded from 350 to 520 nm in 10-nm steps (9.4 nm spectrometer bandpass) to reconstruct the transient difference absorption spectra at different time delays after the pump-pulse excitation. Between each single-shot excitation the cell was moved manually to probe a fresh area. Time traces of the variation in absorbance start to be meaningful at 500 ns. Time windows explored were from 500 ns to 100 μ s (10 ns / point) and from 100 μ s to 2 ms (200 ns / point).’

“

Figure R8: Raw data and global fits with residuals of the kinetic time traces for all the wavelengths for three rsEGFP2 solutions ((a) 50 mM HEPES pH 8, 50 mM NaCl in H_2O , (b) 50 mM HEPES pH 8, 50 mM NaCl in D_2O , (c) 50 mM HEPES pH 8, 50 mM NaCl, 1.25 M $(\text{NH}_4)_2\text{SO}_4$ in H_2O) and (d) for microcrystal suspensions in 100 mM HEPES pH 8, 2.5 M ammonium sulphate.

Supplementary Figure 9: Geometry between pump, probe and irradiation beams at the spectroscopic cell for nanosecond time resolved absorption experiments of (a) protein solutions and (b) microcrystal suspensions.

Minor points:

- p3, ref 8 should actually be ref 9 (photoswitching quantum yields of Dronpa)

The correct reference has been added.

- p4, I would replace ref 18 by Henderson et al., PNAS 2007, 104, 6672 which contains the on and off state structures of mTFP0.7, while ref 18 contains the on state structure of mTFP1

Done.

- p4, third paragraph: "on a time scale from a few ps to (μ s) ms" (there is a 2 ms phase in IrisFP, see Colletier et al., J. Phys. Chem. Lett. 2016, 7, 882)

This was modified as suggested.

- p5, results section and in several other places throughout the manuscript: replace "magenta" by "purple"

'Magenta' has been replaced by 'purple' whenever referring to figure 6. When referring to electron density, we kept 'magenta'.

- p6, second paragraph: *"the 4.4 μ s time constant is mainly characterized by a growth of the positive band at 460 nm - and a decay at 390 nm"*.
"and a decay at 390 nm" has been added.

- p12, second paragraph: *"Chromophore deprotonation occurs with (a characteristic) an average time constant of 759.1 μ s"*.
'average' has been replaced by 'characteristic' and the two deprotonation time constants have been given.

- p18, second paragraph and in the caption of Supplementary Figure 5: *I think you should exchange the 40 and 60 %*.
Well spotted – thank you ! The caption has been modified and 40 and 60% have been exchanged in the text.

- *Figure 5b: The distance between the NE2 of His149 off-like and the phenol oxygen of the chromophore is lacking.*
The lacking distance has been added in fig. 5b.

Reviewer #2 (Remarks to the Author):

The major goal of this paper is to fill in the later (ns) “on” events of the photoswitching behavior of a GFP variant that the group has previously characterized at the ps timescale (in a great paper in Nature Chemistry earlier this year). The major discovery is that the chromophore binding pocket has surprising heterogeneity on this timescale that differ from the traditional “on” structure. This conclusion is reached after analysis of maps created by various manipulations of the structure factors. The observed heterogeneity is likely linked to the protonation of the chromophore, which cannot be revealed directly here, but is nicely inferred through an analysis of distances between heavy atoms. The authors also discover new heterogeneity of the “off state” through the use of phenix.ensemble_refinement.

We thank the referee for his positive comments on our study and the opportunity to clarify the issues he raised.

I have three remaining issues with the paper that would benefit from further clarification:

As with reviewing their last paper in Nature Chemistry, I have some minor issues with the use of extrapolated structure factors and difference refinement. The use of extrapolated structure factors as a discovery/model building tool is appropriate, but given the high occupancy (40-50%), I don't understand why the difference refinement is preferred over refining a traditional multiconformer model (for example, assigning A, B altconfs to the intermediate and C, D, E altconfs to the ground state)

We had indeed first refined the structural 10-ns data according to a traditional multiconformer model and obtained a similar occupancy for the intermediate state (about 50%) as judged by assessing the residual $F_o - F_c$ electron density at the chromophore. However, by using difference refinement, we were able to fade out the ground-state heterogeneity and only concentrated on the new structural features that appear in the 10-ns data, for instance the now unique *off*-like position of His149.

The presence of *Trans2* in the *off* state structure is fascinating. If they subject either previous SFX (from the Nature Chemistry paper) or traditional synchrotron OFF datasets do they detect similar heterogeneity from ensemble refinement or a closer inspection of the map? Similarly, is there any evidence for His149 heterogeneity identified in Figure 5B in previous longer timescale “on state” datasets?

The apparent difference in ground state heterogeneity of earlier XFEL and synchrotron structures and the present work is of course puzzling and was investigated by us early on. Indeed, the additional *trans2* conformation has not been reported in our earlier *off*-state structures determined by traditional cryo-crystallography at a synchrotron (El Khatib et al 2016) and by SFX at room temperature at an XFEL (Coquelle et al 2018). As referee 2 suggests, we had already analyzed those two structures by the same ensemble-refinement approach (see result in Figure R1a and b below) as the one presented in Figures 3b and c so that a low-occupancy *trans2* state could have been identified, if present. Analysis of the *off*-state data collected by cryo-crystallography (Fig. R1a) did not provide evidence of the existence of a *trans2* conformation for the chromophore, nor provided it evidence that an alternative His149 conformation is present. Likewise, analysis of the *off*-state data collected by SFX (Fig. R1b) did not reveal the existence of *trans2* or any alternative His149 conformation (Fig. 3b). Consequently, *trans2* had not been included in the off-state structures determined by SFX (Coquelle et al 2018) and by cryo-crystallography at a synchrotron (El Khatib et al 2016). Figure R1 has been added as an additional supplementary figure (Fig. S8) in the revised version of the manuscript.

Figure R1: Result of ensemble refinements against (a) the *off*-state data in El Khatib et al 2016) starting for the *off*-state model (chromophore 100% *trans1*, PDB code 5DTY) and (b) the *laser_off* data set reported in Coquelle et al 2018 starting for the *off*-state model (PDB code 5O8A) from which the *cis* alternate conformation was omitted so that the chromophore was fully occupied in the *trans1* conformation.

[redacted]

[redacted]

Interestingly, a recent manuscript (Chang et al (2019) JACS) also provides evidence for the existence of a *trans2* chromophore in cryo-crystallographic synchrotron experiments. Illumination at 488 nm of an rsEGFP2 variant with a substituted phenolate ring yielded *trans2* when crystals with a contracted unit cell have been examined, whereas the *trans1* conformation is populated when crystals with a larger unit cell have been used. The authors suggest the tighter packing of proteins in the contracted crystal reduces the flexibility of the chromophore pocket, thereby limiting the volume available for the isomerizing chromophore and favoring the volume-conserving *cis* > *trans2* transition, rather than the volume-consuming *cis* > *trans1* transition. We refer to this manuscript in the Discussion section:

‘Evidence for the existence of a trans2 chromophore also comes from cryo-crystallographic synchrotron experiments, in which illumination at 488 nm of an rsEGFP2 variant with a substituted phenolate ring yielded trans2 when crystals with a contracted unit cell were examined, whereas the trans1 conformation is populated when crystals with a larger unit cell were used (Chang et al 2019). The authors suggest the tighter packing of proteins in the contracted crystal reduces the flexibility of the chromophore pocket, thereby limiting the volume available for the isomerizing chromophore and favoring the volume-conserving cis >

trans2 transition, rather than the volume-consuming *cis* > *trans1* transition. Coincidentally, different illumination conditions were used for the two datasets.’

Our approach to refinement of the 10-ns intermediate-state structure in the current manuscript is based on extrapolated structure factors that only analyse differences between the illuminated and dark structures. Therefore, any *off*-state heterogeneity in the *off*-state structure is cancelled out. In addition, since the 10-ns intermediate structure is occupied at the 50% level in the laser-on structure, we can exclude that it forms from *trans2* that is only occupied at 25% in the laser-off structure before photoswitching with 400 nm light. Yet, it is possible that a fraction of the 10-ns intermediate structure originates from switched *trans 2* conformers. We argue that none of the three main findings in the present work (*cis*-protonated nature of the first ground-state intermediate; interplay between chromophore and protein conformational changes during photoswitching; μ s-ms chromophore deprotonation) are questioned by the presence of *trans2* in addition to *trans1* so that the main conclusions of the manuscript remain valid. [redacted]

I think the authors are missing an opportunity if they leave their discussion of the structural models and the energy diagram of Fig 6 unlinked. Help me understand it by integrating them together in a proposed schematic that unifies what we learned on the ps and ns timescales! In particular, Is the new trans2 related to the model T intermediate of the last paper?

We followed the referee suggestion and modified figure 6 by linking the energy diagram with structural models (see revised figure 6 below). As to model T identified in Coquelle et al 2018, we do not think it is related to *trans2*, since *trans2* was not present in the dark structure of Coquelle et al (see new figure R1b and new supplementary fig. 8b). However, only TR-SFX experiments on the rsEGFP2 V151L mutant can address this issue experimentally. So far, we have collected ultrafast TR-SFX data only on rsEGFP2 V151A (LCLS, LR38, Feb. 2018) but not on V151L. Therefore, we choose not to comment on the link between model T and *trans2* in the revised version of the manuscript.

New Figure 6: Model for the rsEGFP2 *off*-to-*on* photoswitching process. The bold purple bar represents the *laser_on_Δ10ns* intermediate structure determined by TR-SFX. Time constants correspond to those determined by femtosecond (87 ps) and nanosecond-resolved pump-probe UV-visible absorption spectroscopy (5.57, 36.1 and 825 μs) in H₂O solution (50 mM HEPES pH 8, 50 mM NaCl). All three *cis* protonated states (purple bars) are ground-state intermediates, interconversion of which involves rearrangements of the protein matrix (orange arrows). Corresponding structural models for the chromophore and His149 as determined by X-ray crystallography are shown in panels *a*, *b* (this work) and *d*{Coquelle, 2018 #42}. *Trans1* and His149-*off* (light grey) and *trans2* and His149-*supp* (dark grey) of the *trans* protonated *off*-state (*a*, Figure 3d) transitioned to *cis* and His149-*off-like* at 10 ns (*b*, Figure 4b, Figures 5b,e). His149-*off-like* has moved to His149-*on* in the *cis* anionic *on* state (*d*, Figures 5c,f). The structure whose intermediate forms in 5.6 μs remains elusive (*c*).

I review non-anonymously, James Fraser (UCSF). I further suggest that the authors publicly post their work by depositing the manuscript on a preprint server (e.g. BioRxiv) immediately. This will allow the rest of the world have an opportunity to help improve their manuscript prior to a "final version" appearing upon the completion of formal peer review. Additionally, this practice is allowed by Nature Communications.

Reviewer #3 (Remarks to the Author):

This manuscript by Woodhouse et al. aims at addressing the mechanism of reversible photoswitching in GFP-type proteins. It is well established that the process involves both chromophore isomerization and a change of the chromophore protonation state. In recent years, evidence has mounted that reversible photoswitching is a multistep process in either direction, with isomerization in the excited state followed by proton transfer in the ground state.

By combining time-resolved serial crystallography at an X-ray free electron laser and pump-probe time-resolved UV-visible spectroscopy, Weik and co-workers have studied the on-switching of rsEGFP2, i.e., the transition from a non-fluorescent, neutral trans chromophore (off-state) to a fluorescent, anionic cis chromophore (on-state), in great detail. First results on excited-state events on the ps timescale have been published by the authors earlier in Nature Chemistry (ref 20 of the present paper). Here, they have followed up on this work and focused on processes on the nanosecond to microsecond time scales.

The spectroscopic data (collected at 100 ns – 10 ms) suggest that the first ground state intermediate is a neutral (protonated) cis chromophore, which then deprotonates on the μ s time scale to yield the final on-state. The X-ray structure determined 10 ns after the photoswitching laser pulse (laser_on_Δ10ns) shows multiple chromophore conformations and also multiple conformations of other residues, notably His149. After comparing these conformations with those present in the 3D structures of on-state and off-state rsEGFP2 from conventional x-ray crystallography, the authors felt reassured that the first ground state intermediate is indeed a neutral cis chromophore.

The spectroscopic data are straightforward and clearly support the key claim of the paper; i.e., excited-state isomerization followed by ground state deprotonation. The SFX data and their refinement are absolutely state-of-the-art, and a great deal of effort has gone into the refinement of multiple conformations (trans1, trans2, cis). Most pronounced are the changes of His149. Unfortunately, the nature of the proton pathway, which would be important for the mechanism, as well as the observed trans-state heterogeneity have remained elusive.

It is true that our TR-SFX and spectroscopic data do not allow a firm determination of the pathway the proton takes when the chromophore deprotonates in the transition between the presented 10-ns intermediate state and the final anionic on-state. We have now analyzed the structures in greater detail and added the following discussion text:

‘The laser_on_Δ10ns intermediate structure allows proposing three different pathways taken by the proton upon chromophore deprotonation. On the first pathway, the proton is transferred from the chromophore via Thr204OG1 and His149O to the solvent outside the protein as suggested for GFP (Agmon et al). A proton pathway between the solvent and the chromophore has also been suggested to exist in the positive RSFP Padron (Regis Faro et al;Brakemann et al). A second proceeds via His149 and a chain of three water molecules (W377, W158, W144; only W377 is seen in Fig. 5a) connected to His149ND1 to the solvent. A third pathway guides the proton via a Grotthuss mechanism involving W356 and W361 (Fig. 5a and b) to the solvent.’

However, analyzing and comparing the hydrogen-bonding networks in the structures of the 10-ns intermediate state and the final anionic *on*-state allowed us to uncover similarities to the ones between the protonated A form and the anionic B form of wild-type GFP and to hypothesise that ‘*the proton might be transferred from the chromophore via Thr204OG1 and His149O to the solvent outside the protein as suggested for GFP.*’

[redacted]

[redacted]

Interestingly, a recent manuscript (Chang et al (2019) JACS) also provides evidence for the existence of a *trans2* chromophore in cryo-crystallographic synchrotron experiments. Illumination at 488 nm of an rsEGFP2 variant with a substituted phenolate ring yielded *trans2* when crystals with a contracted unit cell have been examined, whereas the *trans1* conformation is populated when crystals with a larger unit cell have been used. The authors suggest the tighter packing of proteins in the contracted crystal reduces the flexibility of the chromophore pocket, thereby limiting the volume available for the isomerizing chromophore and favoring the volume-conserving *cis* > *trans2* transition, rather than the volume-consuming *cis* > *trans1* transition. We refer to this manuscript in the Discussion section:

‘Evidence for the existence of a trans2 chromophore also comes from cryo-crystallographic synchrotron experiments, in which illumination at 488 nm of an rsEGFP2 variant with a substituted phenolate ring yielded trans2 when crystals with a contracted unit cell were examined, whereas the trans1 conformation is populated when crystals with a larger unit cell were used (Chang et al 2019). The authors suggest the tighter packing of proteins in the contracted crystal reduces the flexibility of the chromophore pocket, thereby limiting the volume available for the isomerizing chromophore and favoring the volume-conserving cis > trans2 transition, rather than the volume-consuming cis > trans1 transition. Coincidentally, different illumination conditions were used for the two datasets.’

Our approach to refinement of the 10-ns intermediate-state structure in the current manuscript is based on extrapolated structure factors that only analyse differences between the illuminated and dark structures. Therefore, any *off*-state heterogeneity in the *off*-state structure is cancelled out. In addition, since the 10-ns intermediate structure is occupied at the 50% level in the laser-on structure, we can exclude that it forms from *trans2* that is only occupied at 25% in the laser-off structure before photoswitching with 400 nm light. Yet, it is possible that a fraction of the 10-ns intermediate structure originates from switched *trans 2* conformers. We argue that none of the three main findings in the present work (*cis*-protonated nature of the first ground-state intermediate; interplay between chromophore and protein conformational changes during photoswitching; μ s-ms chromophore deprotonation) are questioned by the presence of *trans2* in addition to *trans1* so that the main conclusions of the manuscript remain valid. [redacted]

The manuscript is a bit difficult to grasp, partially because of the nomenclature.

We would be grateful if the reviewer could point us to instances where we could better explain the nomenclature used so that we can make the manuscript easier to grasp.

Also, clear depictions of the irradiation protocols would be helpful.

We have added in the revised version a supplementary figure (Supplementary figure 9) showing the geometry between pump and probe irradiation beams in the nanosecond time resolved absorption experiments and extended the description of the corresponding irradiation protocols in the materials-and-methods section.

Furthermore, the manuscript could benefit from better working out the new insights gained from the data for the mechanism. The model in Figure 6 does not go beyond a description of the kinetics as measured by spectroscopy. Therefore, the last sentence of the abstract (..establishing a detailed mechanism..) promises more than the present version of the manuscript delivers.

We followed the referee suggestion and modified figure 6 by linking the energy diagram with structural models (see revised figure 6 below). We hope that the proposed photoswitching mechanism is now described in a clearer way.

New Figure 6: Model for the rsEGFP2 *off*-to-*on* photoswitching process. The bold purple bar represents the *laser_on_Δ10ns* intermediate structure determined by TR-SFX. Time constants correspond to those determined by femtosecond (87 ps) and nanosecond-resolved pump-probe UV-visible absorption spectroscopy (5.57, 36.1 and 825 μs) in H₂O solution (50 mM HEPES pH 8, 50 mM NaCl). All three *cis* protonated states (purple bars) are ground-state intermediates, interconversion of which involves rearrangements of the protein matrix (orange arrows). Corresponding structural models for the chromophore and His149 as determined by X-ray crystallography are shown in panels *a*, *b* (this work) and *d*{Coquelle, 2018 #42}. *Trans*1 and His149-*off* (light grey) and *trans*2 and His149-*supp* (dark grey) of the *trans* protonated *off*-state (*a*, Figure 3d) transitioned to *cis* and His149-*off-like* at 10 ns (*b*, Figure 4b, Figures 5b,e). His149-*off-like* has moved to His149-*on* in the *cis* anionic *on* state (*d*, Figures 5c,f). The structure whose intermediate forms in 5.6 μs remains elusive (*c*).

Specific remarks:

Page 6: There are two processes showing a significant KIE, with time constants of 40 and 919 μs, which were population averaged to 759.1 μs. The processes differ by more than an order of magnitude in time. Why should one subsume kinetically so different processes into one process - just for simplicity?

We follow the referee's advice and refrained from calculating an average time constant and specify both time constants in the final scheme (new Figure 6) as characterizing deprotonation. Accordingly, the results section was modified as follows:

“Global fit analysis of kinetic traces for all wavelengths identified three time constants of $5.57 \pm 0.02 \mu\text{s}$, $36.1 \pm 0.1 \mu\text{s}$ and $824.8 \pm 0.3 \mu\text{s}$, respectively. When similar experiments were carried out in D₂O solution (50 mM HEPES pH 8, 50 mM NaCl ; Supplementary Fig. 3), time

evolution also required fitting with three exponential functions, yielding time constants of $5.16 \pm 0.02 \mu\text{s}$, $88.4 \pm 0.2 \mu\text{s}$ and $2041.1 \pm 0.7 \mu\text{s}$. Thus, the first time constant is similar in H_2O and D_2O , but a significant isotope effect is observed for the two others ($k_{\text{H}}/k_{\text{D}} = 2.45$ and 2.47 , respectively) which can be assigned to proton transfer steps. Decay associated spectra (Fig. 2d) show that the $5.57 \mu\text{s}$ time constant in H_2O solution is mainly characterized by a growth of the positive band at 460 nm and a decay at 390 nm. The $36.1 \mu\text{s}$ time constant has some positive and negative contributions characteristic of the band shift observed, while the $825 \mu\text{s}$ time constant is mainly characterized by the respective decay and growth of the 390-nm and 480-nm bands.”

Page 12: I find the second half of the second paragraph (starting with: Transient absorption...) rather unclear. I would recommend elaborating on this issue by using a schematic.

We have modified figure 6 (see above) and extended the end of that paragraph and hope the proposed series of events is now clearer:

‘Transient absorption spectroscopy in H_2O solution evidences a further ground-state process with a time constant of about $5.57 \mu\text{s}$ (no isotope effect) that might correspond to a protein rearrangement during which the off-like conformation of His149 vanishes and the on conformation becomes fully occupied (Fig. 6). Alternatively, the conformational change in His149 could accompany chromophore deprotonation that occurs with characteristic time constants of 36.1 and $825 \mu\text{s}$ (isotope effect) to yield the final cis anionic on-state.’

Page 14, top paragraph: “Woodhouse et al., to be described elsewhere.” All required procedures should be included in this work.

Details given in the materials-and-methods section are sufficient to reproduce microcrystallization. Therefore, we omitted "Woodhouse et al., to be described elsewhere" in the revised version.

Page 15, bottom paragraph: I cannot find a Fig. 8 in Ref 22.

We thank the referee for having spotted this error. The correct figure is Fig. 6, not 8. We have corrected this error in the revised manuscript.

Reviewers' comments:

Reviewer #1 (Remarks to the Author):

In my review of the first version of the manuscript one year ago, I raised the following three main points :

- 1) the questionable novelty of the conclusions in view of the time-resolved spectroscopy literature available at that time ;
- 2) the inconsistency between the heterogeneous off-state structure reported in the present work and the structures reported by the same authors in Coquelle et al., Nat. Chem. 2018, 10, 31 and in El Khatib et al., Sci. Rep. 2016, 6, 1849, which both show a single conformer ;
- 3) the temporal mismatch between the 10-ns intermediate structure and the spectroscopy data starting at 100 ns, that did not support the assignment of the 10-ns structure to the first ground-state photo-intermediate.

Concerning point 1, the authors argue that the context has changed since the recent publication of an extended time-resolved IR spectroscopy study of Dronpa by the Meech group (Laptenok et al., Nat. Chem. 2018, 10, 845), and I agree with them. Time-resolved spectroscopy in the UV-visible and IR spectral ranges currently disagree on the time-scale of trans→cis isomerization in Dronpa by at least three orders of magnitude, which makes time-resolved crystallography studies of ground-state photoswitching intermediates in RSFPs highly relevant. In this new context, a 10-ns rsEGFP2 structure as reported in the present work is certainly of interest.

Concerning point 3, the authors have carried out all the additional time-resolved spectroscopy experiments that were required. All relevant time-scales are now covered and the signal-to-noise ratio has been significantly improved. Interestingly, this additional work revealed a new kinetic phase of 87 ps, assigned to a ground-state process. This means that the 10-ns structure is not the first ground-state photo-intermediate, but rather the second. This is clear in Figure 6 but not in the abstract and in the text. Please delete the adjective « first » where needed.

Point 2 has not been solved and this is in my opinion the main weakness. In the experimental conditions of the present work, the structure of off-state rsEGFP2, which is the initial state of the reaction under study, consists of two trans conformers trans1 and trans2 with respective occupancies of 65 % and 25 % (plus a cis conformer with 10 % occupancy due to incomplete photoswitching). This contrasts with previous off-state structures reported by the same authors (Coquelle et al. 2018 and El Khatib et al. 2016, reprocessed in Figure R1 of the response) which show only a single trans1 conformer.

In response to this point (p.4-6 of response), the authors mention and show (Figure R2) other unpublished TR-SFX data with even more of the trans2 conformer in the off state, that they say will be published soon in another paper. They also cite the recent work by Chang et al. (Chang et al., JACS 2019, 141, 15504) on chlorinated rsEGFP2. The paper by Chang et al. indeed reports two off-state conformers of Cl-rsEGFP2 that resemble the trans1 and trans2 conformers. It seems however that Chang et al. are able to control which conformer is present in their crystals by changing the crystallization conditions, and M. Weik and collaborators have also shown in their previous work that it is possible to prepare rsEGFP2 crystals containing a single conformer. So why not tackle this problem before acquiring more data ?

The authors then argue (p.6 of response) that their approach of analysing differences between the illuminated and dark structures cancels out off-state heterogeneity so that it does not have any impact on the 10-ns structure. It seems to me that this would indeed be the case if only the trans1 conformer would react, but the authors say (p.5 of response) that their unpublished data show that both trans1 and trans2 react to 400-nm excitation. The disappearance of the on-like conformation of His149 from the revised 10-ns structure is also puzzling. I would really feel more comfortable with a single off-state conformer like in the old studies.

Reviewer #3 (Remarks to the Author):

The authors have substantially revised and considerably improved their manuscript in response to the reviewers' remarks. Overall, in my view, they have done an excellent job, I am very pleased

indeed. I still have a few remarks that I would like to bring to the attention of the authors.

1. This point was triggered by a statement on page 13 where the authors reported that, for microcrystals, the time constants changed substantially (4.75 vs 2.6 μ s, 42.9 vs 36.5 μ s and 295 vs 310 μ s) only by using two different fitting programs. Thus, I encourage the authors to do more than running their data through a commercial software package. I am afraid that many readers may come to believe that they should take a figure such as $824.8 \pm 0.3 \mu$ s (4 significant figures!) seriously.

I am painfully aware from own experience that multiexponential fits to inevitably noisy experimental data yield ambiguous results unless the characteristic times are orders of magnitude apart. Typically, when fixing and changing one parameter, the fitting algorithm compensates that with adjustments in others, resulting in flat χ^2 surfaces for this particular parameter. Thus, I am convinced that the quoted errors are unrealistically small ($5.57 \pm 0.02 \mu$ s, $36.1 \pm 0.1 \mu$ s and $824.8 \pm 0.3 \mu$ s), given the rather coarse experimental data (although this is presumably what 'Igor' (i.e., the software) told them). How well does the model describe the data? A plot showing how well the calculated spectra describe the measured spectra would also be helpful.

2. I understand that the two slow timescales (Figure 2) describe kinetic components that display the KIE. However, this is just a formal treatment and it should be made clear that we need not necessarily interpret the fit result in terms of two distinctly different relaxation processes. I am convinced that a stretched exponential, which has been widely used to model slow, cooperative protein dynamics, would be an alternative model describing a "single" kinetic process.

3. In my view, this work provides compelling evidence for the mechanism involving excited-state isomerization followed by ground-state deprotonation and is thus at odds with the interpretations of IR spectra on Dronpa, published in Nature Chemistry. At the end of the article, the authors state the obvious:

If the photoswitching mechanism of RSFPs from Anthozoa is identical to the one from Hydrozoa, our study would disagree with the proposed ground-state isomerization in Dronpa 14,15 and provide unambiguous support to the proposed excited-state isomerisation and ground-state deprotonation¹¹⁻¹³.

Is it reasonable that there should be two distinct mechanisms in two proteins that are so similar in structure or function? Perhaps, the authors can envision reasons for such distinctly different behavior based on structural differences? Or can IR band assignments based on equilibrium structures be erroneous when applied to short-lived, transient structures? I believe that the interested readers would appreciate thoughtful speculations as to how this apparent contradiction can be resolved.

3. This is only a style (or even taste) issue:

From the previous review: The manuscript is a bit difficult to grasp, partially because of the nomenclature.

Response by the authors: We would be grateful if the reviewer could point us to instances where we could better explain the nomenclature used so that we can make the manuscript easier to grasp.

What I tried to convey is that the repeated use of the rather clumsy nomenclature,

"laser_on_ Δ 10ns and laser_off data sets, laser_on_ Δ 10ns intermediate structure, laser_off structure, etc."

makes the text unpleasant to read. Would it not be simpler to explain the preparation of the off-state by pre-illumination and subsequent off-to-on photoactivation (or not) once and then refer to

the '(10-ns) photoproduct' and 'dark' structures?

We are grateful to both referees for their thoughtful, detailed and constructive comments and criticisms. In the following, we reproduce the referee comments in blue. Our responses are in black, including additional figures and references when necessary, and additional text in the revised manuscript is in yellow.

Reviewer #1:

In my review of the first version of the manuscript one year ago, I raised the following three main points :

1) the questionable novelty of the conclusions in view of the time-resolved spectroscopy literature available at that time ; 2) the inconsistency between the heterogeneous off-state structure reported in the present work and the structures reported by the same authors in Coquelle et al., Nat. Chem. 2018, 10, 31 and in El Khatib et al., Sci. Rep. 2016, 6, 1849, which both show a single conformer ; 3) the temporal mismatch between the 10-ns intermediate structure and the spectroscopy data starting at 100 ns, that did not support the assignment of the 10-ns structure to the first ground-state photo-intermediate.

Concerning point 1, the authors argue that the context has changed since the recent publication of an extended time-resolved IR spectroscopy study of Dronpa by the Meech group (Laptenok et al., Nat. Chem. 2018, 10, 845), and I agree with them. Time-resolved spectroscopy in the UV-visible and IR spectral ranges currently disagree on the time-scale of trans→cis isomerization in Dronpa by at least three orders of magnitude, which makes time-resolved crystallography studies of ground-state photoswitching intermediates in RSFPs highly relevant. In this new context, a 10-ns rsEGFP2 structure as reported in the present work is certainly of interest.

We are pleased that this reviewer acknowledges the importance of a 10 ns rsEGFP2 in the new literature context.

Concerning point 3, the authors have carried out all the additional time-resolved spectroscopy experiments that were required. All relevant time-scales are now covered and the signal-to-noise ratio has been significantly improved. Interestingly, this additional work revealed a new kinetic phase of 87 ps, assigned to a ground-state process. This means that the 10-ns structure is not the first ground-state photo-intermediate, but rather the second. This is clear in Figure 6 but not in the abstract and in the text. Please delete the adjective « first » where needed.

We omitted “first” in the abstract.

Point 2 has not been solved and this is in my opinion the main weakness. In the experimental conditions of the present work, the structure of off-state rsEGFP2, which is the initial state of the reaction under study, consists of two trans conformers trans1 and trans2 with respective occupancies of 65 % and 25

% (plus a cis conformer with 10 % occupancy due to incomplete photoswitching). This contrasts with previous off-state structures reported by the same authors (Coquelle et al. 2018 and El Khatib et al. 2016, reprocessed in Figure R1 of the response) which show only a single trans1 conformer. In response to this point (p.4-6 of response), the authors mention and show (Figure R2) other unpublished TR-SFX data with even more of the trans2 conformer in the off state, that they say will be published soon in another paper. They also cite the recent work by Chang et al. (Chang et al., JACS 2019, 141, 15504) on chlorinated rsEGFP2. The paper by Chang et al. indeed reports two off-state conformers of Cl-rsEGFP2 that resemble the trans1 and trans2 conformers. It seems however that Chang et al. are able to control which conformer is present in their crystals by changing the crystallization conditions, and M. Weik and collaborators have also shown in their previous work that it is possible to prepare rsEGFP2 crystals containing a single conformer. So why not tackle this problem before acquiring more data ?

We fully agree with the reviewer in that we have not yet identified the parameter(s) that allow controlling the ratio between trans1 and trans2. Ongoing work aims at identifying these parameters.

The authors then argue (p.6 of response) that their approach of analysing differences between the illuminated and dark structures cancels out off-state heterogeneity so that it does not have any impact on the 10-ns structure. It seems to me that this would indeed be the case if only the trans1 conformer would react, but the authors say (p.5 of response) that their unpublished data show that both trans1 and trans2 react to 400-nm excitation. The disappearance of the on-like conformation of His149 from the revised 10-ns structure is also puzzling. I would really feel more comfortable with a single off-state conformer like in the old studies.

We fully understand the reviewer's point. In fact, first versions of the manuscript written in 2016 only presented one off-state conformer. But we were always left with residual difference electron density that we had not modelled. Rather than omitting it entirely, we felt in the end it would be scientifically most appropriate to report the presence of a minor second off-state conformer, even though we hadn't found yet a way to control its presence or absence. The recent report of Chang et al. (Chang et al., JACS 2019, 141, 15504) corroborates the veracity of our second *off*-state conformation and even presents a likely explanation for its presence in that study.

Reviewer #3:

We have addressed referee 3's main issue concerning the validity of a global fit analysis with a weighted sum of three exponentials by performing a bootstrap estimate of the standard deviation of the set of parameters. The significance of the given precision is now ascertained by residuals devoid of a structure. We added two new figures in the SI (Supplementary figures 3 and 4) that show residuals of the global fit analysis and present results of the bootstrap results. Additional text in the Results and Methods

sections present in detail the additional analysis. All changes made in the manuscript are highlighted in yellow.

The authors have substantially revised and considerably improved their manuscript in response to the reviewers' remarks. Overall, in my view, they have done an excellent job, I am very pleased indeed. I still have a few remarks that I would like to bring to the attention of the authors.

1) This point was triggered by a statement on page 13 where the authors reported that, for microcrystals, the time constants changed substantially (4.75 vs 2.6 μs , 42.9 vs 36.5 μs and 295 vs 310 μs) only by using two different fitting programs. Thus, I encourage the authors to do more than running their data through a commercial software package. I am afraid that many readers may come to believe that they should take a figure such as $824.8 \pm 0.3 \mu\text{s}$ (4 significant figures!) seriously. I am painfully aware from own experience that multiexponential fits to inevitably noisy experimental data yield ambiguous results unless the characteristic times are orders of magnitude apart. Typically, when fixing and changing one parameter, the fitting algorithm compensates that with adjustments in others, resulting in flat χ^2 surfaces for this particular parameter. Thus, I am convinced that the quoted errors are unrealistically small ($5.57 \pm 0.02 \mu\text{s}$, $36.1 \pm 0.1 \mu\text{s}$ and $824.8 \pm 0.3 \mu\text{s}$), given the rather coarse experimental data (although this is presumably what 'Igor' (i.e., the software) told them). How well does the model describe the data? A plot showing how well the calculated spectra describe the measured spectra would also be helpful.

The reviewer is absolutely right, model ambiguity is a known issue in multi-exponential fitting, except for a mono-exponential model. As suggested, we added a new Figure in the supplementary section to show the residuals (Supplementary Fig. S3). Since the residuals do not show any structure, the calculated spectra are indistinguishable from the measured spectra (see Figure R1 for reviewer).

We also took into consideration the comment and questions regarding the significance of the error. We now clearly explain the distinction that should be made between the uncertainty estimated on the model parameters, which is small because it assumes that the actual non-linear parameters estimated are true, and the one that would be estimated from real replicates of the experiment, which of course would be larger as it would embed more sources of variability. Bootstrapping can be considered as a way to check the ambiguity of the model. We now describe this procedure and the motivation for it, and added the results in a new Figure in the SI (Supplementary Fig. S4). Results obtained for 1000 replicates confirm the validity of the time constants reported above and given in the manuscript. We provide below the text that was added in the results section and in the material methods.

Text added in the results section:

Three distinct time windows can be observed for the evolution of transient spectra (Fig. 2a – c) with the existence of two isosbestic points (Fig. 2a, c). A model based on the weighted sum of three exponential decays was chosen to fit the kinetic traces for all wavelengths between 100 ns to 9 ms. The estimated

time constants were $5.57 \pm 0.02 \mu\text{s}$, $36.1 \pm 0.1 \mu\text{s}$ and $824.8 \pm 0.3 \mu\text{s}$, respectively. The residuals show no structure, thus validating the given confidence intervals (Supplementary Fig. S3). However, it is known that standard deviations of the parameter values are optimistic estimates of the confidence on these parameters (since it assumes that the estimated parameters are the true ones)³⁸. Therefore, we performed a bootstrap estimate of the standard deviation of the set of parameters. By bootstrapping, different sets of parameters are reported for each replicated fit. Mean value and confidence can then be estimated from their respective distribution. Results obtained for 1000 replicates are reported in Supplementary Fig. S4 (see details in the Methods section) and confirm the validity of the time constants reported above. Similar experiments were carried out in D₂O solution (50 mM HEPES pD 8, 50 mM NaCl ; Supplementary Fig. S5), and the same model was applied, yielding time constants of $5.16 \pm 0.02 \mu\text{s}$, $88.4 \pm 0.2 \mu\text{s}$ and $2041.1 \pm 0.7 \mu\text{s}$.

New text in the material methods section:

For the analysis of transient absorption spectra, a model based on a weighted sum of three exponential decays was chosen to fit the kinetic traces for all wavelengths (global fit analysis). This was done with Igor Pro and a custom made analysis routine in python 3 using *lmfit* package³¹. The quality of the fit was checked by analyzing the residuals (no structure, Supplementary Fig. S3) and the Levenberg-Marquardt algorithm was used to minimize least-squares error surface ($\chi^2 < 10^3$). The reported values of the standard deviations of the parameters correspond to the diagonal elements of the covariance matrix, which is the inverse of the so-called curvature matrix taken at the minimum of the error. By contrast to linear fitting, this matrix is not constant but a function of the fitting parameters for non-linear modeling. Still, information on standard deviations can be derived but should, however, be interpreted with caution, keeping in mind the assumption that the set of parameters estimated is true. A more reliable estimation of the uncertainty would require replicated estimations of the non-linear parameters on different data sets. These replicated data sets can be generated with bootstrap. Fitting these datasets provides different sets of parameters, whose means and standard deviations can be estimated from their distributions. The model was here fitted on 1000 replicated data sets constructed using the best parameters of the fit and adding replicates of noise (Supplementary Fig. S4). These replicates were obtained from the residuals (difference between the original and reconstructed data) randomly shuffling 25% of the values. Figures were made using *Igor Pro* and *Seaborn and matplotlib* python 3 package.

Supplementary Figure 3: Raw data (points in lower part of the panels) and global fit analysis (lines in lower part of the panels) with residuals (res; upper part of the panels) of the kinetic traces for 19 wavelengths (different colors, between 350 and 520 nm, 10 nm steps) with a weighted sum of three exponential decays for three rsEGFP2 solutions. (a) 50 mM HEPES pH 8, 50 mM NaCl in H₂O, (b) 50 mM HEPES pH 8, 50 mM NaCl in D₂O, (c) 50 mM HEPES pH 8, 50 mM NaCl, 1.25 M (NH₄)₂SO₄ in H₂O and (d) rsEGFP2 microcrystals in 100 mM HEPES pH 8, 2.5 M ammonium sulphate.

Supplementary Figure 4: Distribution of the three time constants retrieved for the 1000 replicates obtained by the bootstrapping procedure for rsEGFP2 solutions: (a, e, i) 50 mM HEPES pH 8, 50 mM NaCl in H₂O; (b, f, j) 50 mM HEPES pD 8, 50 mM NaCl in D₂O; (c, g, k) 50 mM HEPES pH 8, 50 mM NaCl, 1.25 M (NH₄)₂SO₄ in H₂O; (d, h, l) rsEGFP2 microcrystals in 100 mM HEPES pH 8, 2.5 M ammonium sulphate. Estimates of average μ and standard deviation σ are provided in microseconds.

Figure R1. Comparison of the transient absorption spectra for different time windows of the raw data (left panel) and the calculated ones (right panel) using the three-exponential model ($5.57 \pm 0.02 \mu\text{s}$, $36.1 \pm 0.1 \mu\text{s}$ and $824.8 \pm 0.3 \mu\text{s}$) for rsEGFP2 in H_2O .